



# Synergy of Active- and Passive Remote Sensing: An Approach to Reconstruct Three-Dimensional Cloud Macro- and Microphysics

Lucas Höppler[1], Felix Gödde[1], Manuel Gutleben[2], Tobias Kölling[1], Bernhard Mayer[1], and Tobias Zinner[1]

[1]Meteorological Institute, Ludwig-Maximilians-Universität München (LMU), Munich, Germany
[2]Institute for Atmospheric Physics, German Aerospace Center (DLR), Oberpfaffenhofen, Germany

**Correspondence:** lucas.hoeppler@physik.uni-muenchen.de

**Abstract.** This paper presents a method to retrieve three-dimensional cumulus cloud macro- and microphysics measured by remote sensing instruments on the German research aircraft HALO. This is achieved by combining our hyper-spectral pushbroom spectrometer specMACS with active and passive remote sensing instruments, such as a lidar, a microwave radiometer, a radar and dropsondes. Two-dimensional cloud information such as cloud size, optical thickness, effective radius and thermodynamic

phase are retrieved by specMACS with established remote sensing methods. Information of the other active and passive remote sensing instruments with a smaller field-of-view are mapped to the wider specMACS swath following Barker et al. (2011). The combination of specMACS with passive and active remote sensing quantities, for example, the Cloud Top Height from lidar measurements, allows new possibilities: three-dimensional cloud macrophysics can be reconstructed. Applying a sub-adiabatic microphysical model constrained with measurements allows to extend the measured quantities to a three-dimensional representa-

tion of microphysics. A consistency check by means of a three-dimensional radiative transfer simulation of the specMACS observations of these derived three-dimensional cloud fields shows good agreement.

## 1   Introduction

Clouds cover on the average about two thirds of our planet (Stubenrauch et al., 2013). They regulate our weather, the hydro-

logical cycle and influence the climate system (Boucher et al., 2013). Cloud processes act from micrometer scale up to scales of thousands of kilometers. Hence, it is impossible to resolve all in one model (Grabowski et al., 2019). For example, the cloud droplet effective radius having a typical size of $10\,\mu m$ interacts at scales by orders of magnitude smaller than those of global climate and cloud models. Current global cloud resolving models, for example, the Nonhydrostatic Icosahedral Atmospheric Model (NICAM, Satoh et al. (2008)), have a horizontal resolution as small as $3.5\,km$ (Putman and Suarez, 2011) and climate

models have a typical horizontal resolution of about $100\,km$ (Taylor et al., 2012). These models cannot resolve shallow cumulus clouds, although, these clouds create one of the largest model spreads in climate sensitivity (Vial et al., 2018).



In 2013, the Intergovernmental Panel on Climate Change stated once again that clouds contribute to the largest uncertainties in our Earth's changing energy budget (Boucher et al., 2013). Clouds influence radiation and, vice versa, radiation influences clouds (Tao et al., 1996). Incoming solar radiation is partially reflected by low-level clouds. The pattern of shadows on the ground acts back on the atmosphere and even cause shallow cumulus cloud streets (Jakub and Mayer, 2017). Outgoing ther-

mal radiation contributes to clustering of shallow cumulus clouds (Klinger et al., 2017). Without discussion, clouds are highly complex physical phenomena with complex macrophysical structure and complex microphysical behavior.

There are uncertainties in mid-latitudes. The forecast of the North Atlantic cyclone development contains uncertainties due to diabatic processes (Rodwell et al., 2013). Diabatic processes, such as latent heat release or radiative heating and cooling,

modify the potential vorticity of cyclones and thus their dynamics (Chagnon and Gray (2015), Chagnon et al. (2013)). In an idealized simulation, Schäfer and Voigt (2018) performed baroclinic life cycle simulations in a global atmosphere model with and without radiation. They showed that radiative diabatic processes reduce the peak kinetic energy of idealized midlatitude cyclones up to $50\%$.

There are also uncertainties in the tropical regions. Radiative cloud interaction favors cloud aggregation which plays a role in large-scale convergence (e.g., Bony et al. (2016), Hartmann et al. (1984)). Especially maritime boundary layer clouds, such as shallow cumulus clouds, seem to cause the present major uncertainties in the climate and weather models (e.g., Vial et al. (2018), Schnitt et al. (2017), Boucher et al. (2013), Vial et al. (2013), Bony and Dufresne (2005)). Therefore, climate models and global weather forecast models need to be improved with respect to a proper representation of microphysical and macro-

physical cloud properties and their diabatic effects.

The aim of the NARVAL-II campaign (Next-Generation Aircraft Remote Sensing for Validation II, Stevens et al. (2019)) was to study clouds and circulation over the western subtropical Atlantic Ocean. Datasets obtained during the campaign are used to both develop and verify our three-dimensional macro- and microphysical cloud retrieval. During NARVAL-II campaign, the

German research aircraft HALO (High Altitude Long-Range Research Aircraft, see e.g. Krautstrunk and Giez (2012) or Ziereis and Glässer (2006)) was equipped with a carefully selected payload of several downward-looking instruments complementing each other. Ten research flights were performed around the Atlantic inter-tropical convergence zone during August 2016.

Approaches to combine remote sensing instruments with each other exist for years: A radar and a lidar were combined to

retrieve cirrus cloud particle sizes and other ice microphysics (e.g., Okamoto et al. (2010), Delanoë and Hogan (2010), Delanoë and Hogan (2008), Donovan and van Lammeren (2001), Intrieri et al. (1993)). A space born radar, a lidar and several infrared radiometers were combined to retrieve ice cloud properties (Delanoë and Hogan, 2010). A microwave radiometer was combined with a cloud radar for the liquid water content retrieval (e.g., Frisch et al., 1995). Wolf et al. (2019) combined a radar, a lidar, a microwave radiometer and a one-dimensional spectrometer to derive the cloud droplet number concentration

by means of three different methods for shallow cumulus clouds. However, this concentration varied strongly within one cloud



scene and depended on the method. Wolf et al. (2019) also mentioned that these values are highly affected by shadows on cloud edges which could not have been taken into account (see also e.g., Marshak et al. (2006)). Shadow effects can be eliminated in our case by combining the lidar with our pushbroom spectrometer specMACS (spectrometer of the Munich Aerosol Cloud Scanner, Ewald et al. (2016)) which was done following the approach of Barker et al. (2011). Barker et al. (2011) described

for the first time a method to combine one-dimensional, satellite-based active remote sensing data with the pushbroom spectroradiometer MODIS aboard the Aqua and Terra satellites. This algorithm is the cornerstone for the combination of passive and active instruments in our research.

The hyper-spectral pushbroom spectrometer specMACS allows the determination of horizontal two-dimensional informa-

tion such as cloud fraction and cloud size distribution (see Appendix B), stereoscopic cloud top height (Kölling et al., 2019), optical thickness, effective radius and the thermodynamic phase. This retrieved specMACS data was then combined with other passive and active remote sensing instruments in form of a microwave radiometer, a lidar and a radar, following the approach of Barker et al. (2011). The cloud top height from the WALES lidar (Wirth et al., 2009) and the cloud bottom height calculated from dropsonde data create the possibility to reconstruct single-layer cumulus clouds in all three dimensions. Using the liquid

water path from the HAMP radiometer (Mech et al., 2014) derived by Jacob et al. (2019a) and the cloud droplet effective radii retrieved from specMACS as constraints for a simple microphysical model allows us to resolve the three-dimensional cloud microphysics. In addition, the radar is used to determine multiple cloud layers.

The purpose of this study is to reconstruct three-dimensional macro- and microphysical cloud properties consistent with all

available remote sensing observations for the observed trade wind cumulus and stratocumulus fields. These datasets will form the basis for a systematic analysis of radiative diabatic effects of these cloud fields.

Section 2 will give an overview of the NARVAL-II campaign, the instrumentation and the radiative transfer software libRadtran. Section 3 describes the theoretical cloud macro- and microphysical properties and the simple sub-adiabatic microphysical

model used to reconstruct the three-dimensional cloud microphysics. Section 4 describes how passive and active remote sensing instruments are combined, which data needs to be excluded and presents our three-dimensional cloud reconstructions. Finally, Section 5 discusses and concludes our research.

## 2   Instruments, Alignment and Radiative Transfer Software

The NARVAL-II campaign took place close to Barbados for four weeks in August 2016 (Stevens et al., 2019). Ten research

flights were conducted with the German research aircraft HALO. The NARVAL-II campaign is closely related to the multi-aircraft campaign called NAWDEX (North Atlantic Waveguide and Downstream Impact Experiment) which took place in Iceland during autumn of the same year (Schäfler et al., 2018). HALO had in both campaigns the same remote sensing instruments aboard with their sensor eyes looking downwards. The instruments are: the pushbroom spectrometer specMACS (Ewald





et al., 2016), the one-dimensional spectrometer SMART (Wendisch et al., 2001), the lidar system WALES (Wirth et al., 2009), the radiometer and cloud radar HAMP (Mech et al., 2014), the HALO internal sensor system BAHAMAS and dropsondes. They will be described in the following.

## 2.1 specMACS

The spectrometer of the Munich Aerosol Cloud Scanner (specMACS) was developed at the Ludwig-Maximilians-Universität of Munich (LMU) and used to derive optical and macro- and microphysical cloud properties (Ewald et al., 2016). It is the main instrument allowing to map nadir remote sensing information with a limited field-of-view to the wider image swath for a synergistic retrieval. It is a hyper-spectral pushbroom spectrometer based on two line cameras measuring radiation from 400 to 2500 nm with a spectral bandwidth ranging between 2.5 and 12 nm. The spatial dimension allows to resolve the two-

dimensional cloud structure. The across-track field-of-view is 32.7° for the visible near-infrared (VNIR) camera and 35.5° for the shortwave infrared (SWIR) camera produced by SPECIM (Specim, Spectral Imaging Ltd.). VNIR has 1312 and SWIR 320 spatial pixel on its swath. Additionally, specMACS has one two-dimensional RGB camera with a larger field-of-view. The temporal resolution during the campaigns was 30 Hz and the instantaneous field-of-view is across-track about 1.4 mrad for the VNIR and about 3.8 mrad for the SWIR. The along-track instantaneous field-of-view is about 2 mrad for both cameras.

This allows, for example, to resolve 30 m surface features along the across-track line of 8.7 km from 15 km altitude. Recently, specMACS got extended by a wide-field-of-view polarization imager.

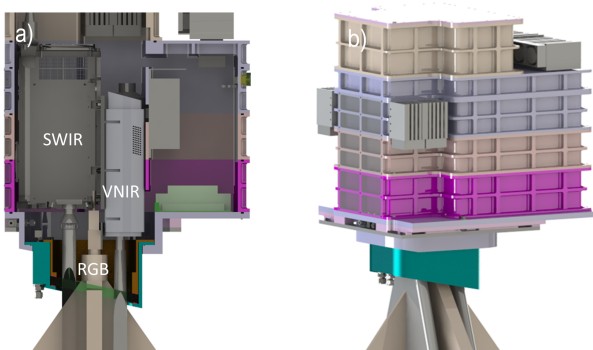

**Figure 1.** The specMACS in HALO configuration: a) shows the open pressurized containment including the cameras as installed in the boiler room of HALO. The two hyper-spectral line cameras SWIR and VNIR as well as a two-dimensional RGB camera are marked. b) shows the closed pressurized containment. Illustrations are taken with permission from *enviscope GmbH*.

## 2.2 HAMP

The HALO Microwave Package (HAMP) was used to retrieve the LWP (Liquid Water Path) among other quantities. It consists of passive microwave radiometers with 26 channels with frequencies ranging between 20 and 183 GHz and an active 35.5 GHz





cloud radar. Seven K-band channels ranging from about 22 to 31 GHz and the 90 GHz channel are used to derive the LWP. This is possible since the brightness temperature increases at about 22.2 GHz with increasing liquid water. The Passive and Active Microwave Transfer code PAMTRA (Maahn and Löhnert, 2017) and dropsonde profiles are used to create simulated HAMP measurements (Jacob et al., 2019a). Comparing simulations with measurements using a linear regression model (Mech et al.,

2007) allows the LWP determination. The LWP values have an absolute accuracy of about $20\,\mathrm{g\,m^{-2}}$ for LWP values below $100\,\mathrm{g\,m^{-2}}$ and an accuracy of $20\,\%$ for values above (Jacob et al., 2019a). The field-of-view of the microwave radiometer K-Band has a diameter of $5.0°$ and the temporal resolution is $1\,\mathrm{s}$.

## 2.3   WALES

The airborne multi-wavelength WALES lidar instrument (WAter vapor Lidar Experiment in Space, Wirth et al. (2009)) is a

combined water vapor Differential Absorption (DIAL) and High Spectral Resolution Lidar system (HSRL, Esselborn et al. (2008)). It allows direct and simultaneous measurements of water vapor mixing ratios and aerosol optical properties from aircraft to ground level. Particle extinction at $532\,\mathrm{nm}$ as well as particle linear depolarization and particle backscatter at 532 and $1064\,\mathrm{nm}$ can be retrieved. The high vertical resolution of $15\,\mathrm{m}$ allows detecting cloud top heights with high vertical accuracy (Gutleben et al., 2019). WALES data with $5\,\mathrm{Hz}$ temporal resolution corresponding to a horizontal resolution of approximately

$170\,\mathrm{m}$ at typical aircraft speed are used in this study.

## 2.4   Dropsondes, BAHAMAS and External Quantities

Dropsondes are radiosondes that are dropped from an aircraft measuring temperature, pressure, relative humidity and horizontal wind velocity and direction. The dropsonde system (HALO-DS, Hock and Franklin (1999)) of the German Aerospace Center and the corresponding dropsondes are used to get the atmospheric temperature and pressure profiles as well as the

surface wind velocity for our radiative transfer simulations. Moreover, the surface temperature and the dew point are used to calculate the Lifted Condensation Level (LCL) and the corresponding Cloud Bottom Height $z_{cbh}$. The dropsondes were released at irregular time steps, ranging from several minutes to hours, depending on the synoptic situation and flight security. In each flight between 10 and 50 dropsondes were released resulting in an estimated spatial resolution ranging roughly from 50 to $400\,\mathrm{km}$.

The Basic Halo Measurement and Sensor System (BAHAMAS) provided us highly accurate aircraft position and orientation data with a temporal resolution of $100\,\mathrm{Hz}$ such as velocity, altitude, location, as well as the principal axes roll, pitch and yaw. These data are necessary, for example, for the determination of the CTH (Cloud Top Height), transformations of camera coordinates into Cartesian coordinates and radiative transfer simulations.





External quantities such as the Solar Zenith Angle (SZA) and Solar Azimuth Angle (AZI) are important for the retrieval of the optical thickness and for developing a shadow mask. Both angles are calculated using the *PyEphem package* which is known for high-precision astronomy calculations (Rhodes, 2011).

## 2.5  Coordinate Systems

5  The WGS-84 (World Geodetic System 1984) is used as the *earth coordinate system* which includes the flattening of the earth (see e.g. technical report, DMA (1991)). It is used as reference system for the GPS (Global Positioning System) and is, for example, the standard global reference system of the U. S. Department of Defense (Groten et al., 1988). The *reference coordinate system* is a Cartesian coordinate system with NED (North-East-Down) coordinates using the WGS-84 surface as as fundamental plane. It is located at height $0\,\mathrm{m}$ on the WGS-84 ellipsoid at arbitrary coordinates (e.g. at $19°$ north and $49°$ west 10  for NARVAL-II). The north axis is parallel to the longitudes and the east axis is parallel to the latitudes.

The *local reference coordinate system* is also a *reference coordinate system* in Cartesian coordinates. The only difference is that this one is located relatively close to the observer which is in our case the HALO position. It is used for reconstruction of the three-dimensional cloud macro- and microphysics. Data of WALES, specMACS and HAMP were transformed into our 15  *reference* or *local reference coordinate system* as described above. The aircraft HALO has its own coordinate system. Inside the HALO, the coordinate systems of BAHAMAS, WALES as well as the VNIR and SWIR cameras are located.

## 2.6  Sensor Positions

The specMACS could not be aligned perfectly inside the HALO aircraft because of time constrains. The viewing direction of the SWIR camera was aligned about $2.6°$ towards the HALO aircraft nose and the center pixel of the SWIR camera was not 20  oriented perfectly vertical downwards along the yaw axis of the aircraft. The WALES lidar instead looks as good as technically possible vertically downwards along the yaw axis of the aircraft. That means that the SWIR camera will see a cloud earlier or, in rare cases, later than the WALES lidar by a time offset of $\Delta t_{cth}$ depending on the CTH. Figure 2 shows schematically the alignment between WALES and the SWIR camera inside specMACS. In the following, the angular offset between the instruments will be corrected. A not-corrected angle offset in viewing direction might influence the results particularly.

Firstly, the time offset needs to be found: We calculate vector $\boldsymbol{x}$ which is the projection of vector $\boldsymbol{p}$ defined as

$$\boldsymbol{p} = \hat{\boldsymbol{e}}_{\mathrm{WALES}}\left(z_{plane} - z_{cth}\right),\tag{1}$$

with $\hat{e}_{\mathrm{WALES}}$ as the WALES viewing vector onto the SWIR viewing direction $\hat{e}_{\mathrm{SWIR}}$

$$\boldsymbol{x} = \boldsymbol{O}_{\mathrm{SWIR}} + \hat{\boldsymbol{e}}_{\mathrm{SWIR}} \cdot \left(\boldsymbol{p} \cdot \hat{\boldsymbol{e}}_{\mathrm{SWIR}}\right).\tag{2}$$

30  As can be seen in Equation 1, vector $\boldsymbol{p}$ is determined using the CTH derived from the WALES lidar. This vector ends in point $\boldsymbol{P}$. The vector $\boldsymbol{x}$ ends in point $\boldsymbol{X}$. Now we use the x-component of the differential vector between $\boldsymbol{P}$ and $\boldsymbol{X}$ which indicates





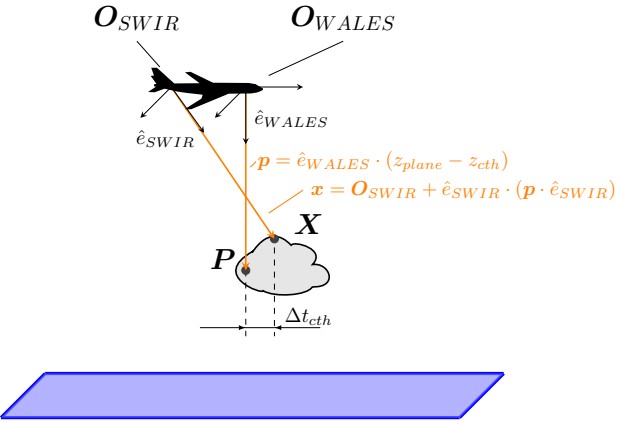

**Figure 2.** Illustration of the alignment of the SWIR camera which looks $2.6°$ towards the HALO nose. $O_{\mathrm{SWIR}}$ denotes the origin of the SWIR coordinate system and $O_{\mathrm{WALES}}$ denotes the origin of the WALES coorindate system. WALES looks directly downwards. The SWIR camera will see a cloud by an time offset of $\Delta t_{cth}$ earlier than WALES when the cloud is sufficiently far away. This time offset depends also on the CTH. The higher the cloud, the smaller the time offset.

the horizontal distance between the two points. The temporal offset can then be calculated by dividing this x-component by the aircraft velocity.

Secondly, relative angular offsets between the instruments were identified by correlating the SWIR radiance measured in
the nadir pixels with the BSR (BackScattering Ratio) at $532\,\mathrm{nm}$ measured by the WALES lidar for different rotation angles. Having found the best correlation, we can rotate the SWIR camera system with the rotation matrix $\mathcal{R}$

$$\mathcal{R} \equiv \mathcal{R}_z(\alpha) \cdot \mathcal{R}_y(\beta) \cdot \mathcal{R}_x(\gamma), \tag{3}$$

where $\alpha$, $\beta$ and $\gamma$ describe the rotation around the $z$, $y$ and $x$ axis in the SWIR coordinate system, respectively. $\alpha$ denotes the yaw angle offset, $\beta$ denotes the pitch angle offset and $\gamma$ denotes the roll angle offset. What is not taken into account is the
bending of the whole aircraft fuselage, but normally this is a small effect within such a short aircraft as HALO. Therefore, this can be neglected.

### 2.7   Radiative Transfer Calculations

The radiance arriving at the observer for different wavelengths is modeled using either a plane-parallel or a three-dimensional atmosphere. In case of a plane-parallel atmosphere, the one-dimensional radiative transfer solver DISORT (Discrete Ordinates
Radiative Transfer Program for a Multi-Layered Plane-Parallel Medium, Stamnes et al. (1988)) is used. In case of a three-dimensional atmosphere, the Monte Carlo code MYSTIC (Mayer, 2009) is used. Both are included in the radiative transfer





library libRadtran (www.libradtran.org, Mayer and Kylling (2005); Emde et al. (2016)).

As libRadtran input we used the closest dropsonde profile and extended it above the aircraft with the profile of the *tropical atmosphere* as defined by (Anderson et al., 1986). A background aerosol with a surface visibility of $50\,\mathrm{km}$ was defined and

the proper sun-earth distance is used. The reflectance of the ocean is calculated according to the Bidirectional Reflectance Distribution Function (BRDF) following Cox and Munk (1954) using the wind velocity of the nearest dropsonde. Mie scattering (Mie, 1908) is used for clouds. The REPTRAN band parameterization (Gasteiger et al. (2014), Buehler et al. (2010)) with medium resolution ($5\,\mathrm{cm}^{-1}$) is used for molecular absorption.

## 3   Theory: Cloud Physics

### 3.1   Macrophysics

The CBH (Cloud Bottom Height $z_{cbh}$), the CTH (Cloud Top Height $z_{cth}$) and the horizontal extension describe the macrophysics of clouds. Figure 3 shows one cloud scene example taken at the NARVAL-II campaign on 19th August 2016. The specMACS radiance at $1553\,\mathrm{nm}$ is shown on the left. White and light gray areas mark the clouds and the ocean is shown in dark gray colors. The reflection of the sun in the ocean is also slightly visible. On the right, the cloudmask is shown. White

areas mark clouds and blue areas mark the ocean. The horizontal cloud dimensions are derived from this cloudmask (see Appendix B).

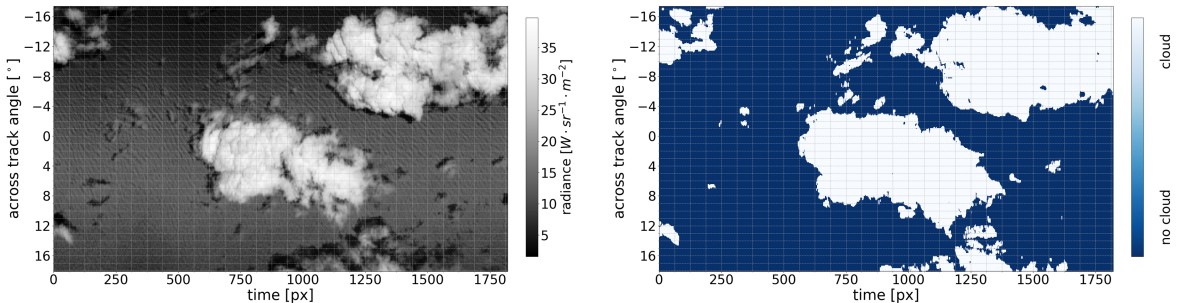

**Figure 3.** The radiance at $1553\,\mathrm{nm}$ is shown on the left and the cloudmask is shown on the right for one cloud example at the NARVAL-II campaign. The aircraft flies in this example to the right.

Typical cumulus clouds form due to rising warm and moist air parcels that cool with height adiabatically. The temperature of the air parcels decreases with height following the Dry Adiabatic Lapse Rate of $9.8\,\mathrm{K\,km}^{-1}$. Similarly, the Dew Point

Temperature decreases with height with a rate of $1.8\,\mathrm{K\,km}^{-1}$ at low altitudes. At some height, called Convective Condensation



Level, the relative humidity $f$ exceeds $100\,\%$ and condensation starts. In this case, the temperature of the ambient air $T$ is equal to the Dew Point Temperature $\tau_d$. Thus, the CBH $z_{cbh}$ in units km, can be calculated with

$$z_{cbh} \approx \frac{T(0)[\text{in }K] - \tau_d(0)[\text{in }K]}{8\,K\,km^{-1}}, \tag{4}$$

where $T(0)$ is the ambient temperature and $\tau_d(0)$ is the dew point temperature at sea surface level (e.g., Wallace and Hobbs,

2006). The measured humidity profile of the dropsonde is not used since the dropsonde seldomly flies through a cloud.

According to Gutleben et al. (2019), the along-track cloud top height $z_{cth}$ is determined using a threshold of 20 in vertical profiles of backscatter ratio. The Cloud Geometrical Thickness $z_{cgt}$ can then be calculated using

$$z_{cgt} = z_{cth} - z_{cbh}. \tag{5}$$

**3.2  Microphysics**

Cloud microphysics is described by the Liquid Water Content (LWC), the effective radius $r_{\text{eff}}$, the cloud optical thickness $\tau$, the thermodynamic phase $I_p$, the Cloud Droplet Number Concentration $N$ and the cloud droplet number density $N(a)$.

The effective radius $r_{\text{eff}}$ of cloud droplets was defined by Hansen and Travis (1974) as an extinction-weighted mean of the

size distribution of the droplets, i.e., the ratio of the third to second moment of the size distribution

$$r_{\text{reff}}(z_c) = \frac{\int_{a_1}^{a_2} a(z_c) \cdot \pi a(z_c)^2 N(a) da}{\int_{a_1}^{a_2} \pi a(z_c)^2 N(a) da}, \tag{6}$$

which is often expressed in $\mu$m. $z_c$ is the height above the CBH, $N(a)$ is the cloud droplet number density and $a$ is the radius of the cloud droplets.

The cloud optical thickness $\tau_c$ can be expressed as the the integrated Extinction Efficiency $Q_{ext}$ between the heights $z_{c1}$ and $z_{c2}$ in a plane-parallel atmosphere approximation. Hansen and Travis (1974), for example, expressed the optical thickness as

$$\tau(z_{c1}, z_{c2}) = \int\limits_{z_{c1}}^{z_{c2}} \int\limits_{0}^{\infty} Q_{ext}(z_c) \pi a^2 N(a) \cdot da \, dz_c. \tag{7}$$

The Liquid Water Content is the amount of liquid water per unit volume of air and often expressed in $\text{g}\,\text{m}^{-3}$. The LWP (Liquid Water Path) is defined as the integral of the Liquid Water Content over the vertical height of the cloud $z_c$, from cloud

bottom $z_{cbh}$ to cloud top $z_{cth}$ and given in $\text{g}\,\text{m}^{-2}$. It can be expressed as

$$LWP = \int\limits_{z_{cbh}}^{z_{cth}} LWC(z_c) \cdot dz_c = \int\limits_{z_{cbh}}^{z_{cth}} \frac{4\pi}{3} \rho_{lw} \int\limits_{0}^{\infty} a(z_c)^3 N(a) da dz_c, \tag{8}$$





with the density of liquid water $\rho_{lw}$.

The thermodynamic phase can be derived using the approach of Jäkel et al. (2013)

$$I_p = \frac{L_{1700} - L_{1553}}{I_{1700}}, \tag{9}$$

where $L_{1700}$ is the radiance at $1700\,\mathrm{nm}$ and $L_{1553}$ is the radiance at $1553\,\mathrm{nm}$. A positive slope $I_p$ shows that ice clouds are observed and a negative slope $I_p$ means that water clouds are observed. This approach is possible since electromagnetic radiation in the wavelength range between $1553$ and $1700\,\mathrm{nm}$ is absorbed stronger by ice than by liquid water (e.g. Pilewskie and Twomey, 1987).

### 3.3 Adiabatic Theory

Liquid Water Content and effective radius $r_{eff}$ are both a function of cloud height $z_c$. It is interesting to know how both develop with height $z_c$. Air rises due to thermal updrafts or lifting and starts to condensate when the rate of condensation is larger than the rate of evaporation. The air gets *saturated* and water condenses on condensation nuclei. Droplet growth by condensation happens when the radii are below $18\,\mu\mathrm{m}$ and no ice exists (Rogers, 1976). This droplet growth can be described by an adiabatic model. Similar adiabatic models have already been used by, for example, Brenguier et al. (2000).

The adiabatic Liquid Water Content is the maximum amount of liquid water that can be contained in a specific volume. Thus, we can express it with the density of condensed water $\rho_{H_2O}$ using

$$LWC_{ad} \equiv \frac{m_{lw}}{V_{air}} = \rho_{H_2O}, \tag{10}$$

where $m_{lw}$ is the mass of liquid water and $V_{air}$ is the specific volume of air. The partial pressure of the condensing water $e_{con}$ must be the difference between the partial pressure of the water vapor $e$ at a given height and the saturation pressure $e_{sat}$

$$\Delta e_{con}(T_c(z_c)) = e(T_c(z_c)) - e_{sat}(T_c(z_c)), \tag{11}$$

where $z_c$ is the height above the cloud bottom and $T_c(z_c)$ the cloud temperature profile. This difference increases with height after the Lifted Condensation Level is reached (see Figure 4).

Using the ideal gas law, the adiabatic Liquid Water Content (Equation 10) becomes

$$LWC_{ad}(z_c) = \frac{e(T_c(z_c)) - e_{sat}(T_c(z_c))}{R_v \cdot T_c(z_c)}, \tag{12}$$

where $R_v$ is the specific gas constant for water vapor. Furthermore, the ideal gas law states inside the cloud

$$e(T_c(z_c)) = \rho_v(T_c(z_c)) \cdot R_v \cdot T_c(z_c), \tag{13}$$

where $\rho_v$ is the density of water vapor. The density of water vapor $\rho_v$ can be expressed with the specific humidity $q$. With the help of the ideal gas law for the density of air $\rho_{air}$, the partial pressure of water vapor $e(T_c(z_c))$ can be described as

$$e(T_c(z_c)) = \frac{q}{1-q} \cdot p_{air}(T_c(z_c)) \cdot \frac{1}{0.622}, \tag{14}$$





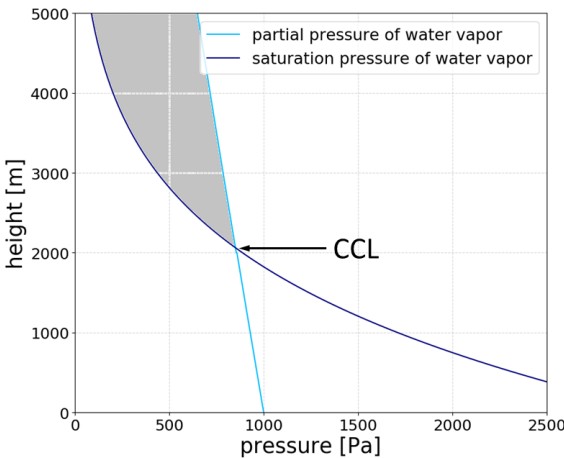

**Figure 4.** Schematic showing the partial pressure of water vapor $e$ (light blue line) and the saturation pressure $e_{sat}$ (dark blue line) decreasing with height. Condensation starts when the Lifted Condensation Level is reached, i.e., when saturation pressure is equal the partial pressure of water vapor. The gray area shows the maximum condensed water $e_{con}$. Droplets will grow as long as the partial pressure is larger than the saturation pressure. This schematic is initialized with $25\,°C$ surface temperature and a surface partial pressure for water vapor of $10\,hPa$.

with the air pressure $p_{air}(T_c(z_c))$. Inserting this Equation into Equation 12 results in:

$$LWC_{ad}(z_c) = \left[\frac{q}{1-q} \cdot p_{air}(T_c(z_c)) \cdot \frac{1}{0.622} - e_{sat}(T_c(z_c))\right] \Big/ R_v \cdot T_c(z_c) \tag{15}$$

The change of saturation pressure with cloud height $e_{sat}(T_c(z_c))$, the change of temperature inside the cloud $T_c(z_c)$ and the change of air pressure with cloud height $p_{air}(T_c(z_c))$ can be calculated using the Clausius-Clapeyron equation, the moist
adiabatic lapse rate and the barometric formula, respectively (see Appendix A).

The theoretical adiabatic Liquid Water Content value is usually not measured but a lower one since, for example, entrainment processes play a role. Morton et al. (1956) defines entrainment as the incorporation of ambient air into a localized circulation. This mixing of environmental air into organized air clusters leads to the evaporation of cloud droplets and also can reduce the
cloud droplet radii. Therefore, we get a measured Liquid Water Content $LWC_m$ which is less or equal than the theoretically calculated adiabatic Liquid Water Content $LWC_{ad}$

$$LWC_m(z_c) = a \cdot LWC_{ad}(z_c). \tag{16}$$

The parameter $a \in [0,1]$ is the degree of entrainment and shows how strong the measured Liquid Water Content differs from the theoretical value, i.e., how strong the entrainment is. The left part of Figure 5 shows some Liquid Water Content profiles
for different entrainment factors $a$ according to Equations 16 and 15. The closer the factor $a$ to 1, the weaker the entrainment.





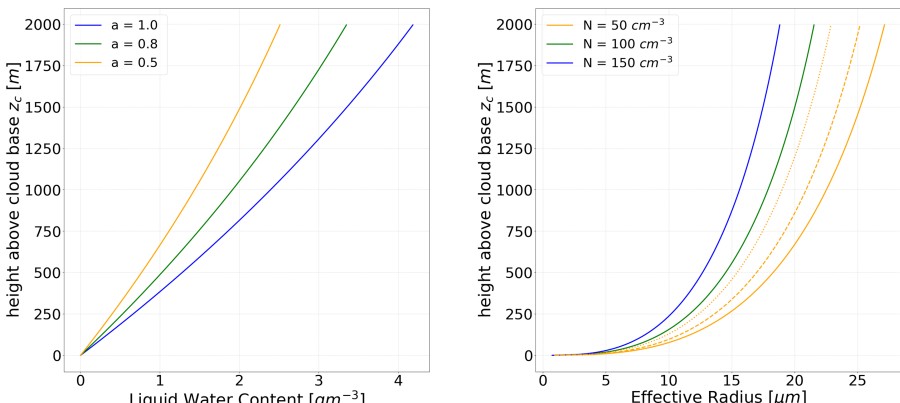

**Figure 5.** The Liquid Water Content for different entrainment factors $a$ is shown on the left part. The higher the entrainment factor, the stronger increases the LWC with height above cloud base $z_{cbh}$. The error bars are not shown since the relative error is only about $3\%$. Three effective radius profiles for three different *cloud droplet number concentrations* are shown on the right part. The higher the *cloud droplet number concentration*, the smaller the *effective radius*. The dashed and dotted orange lines show the effective radius for a Cloud Droplet Number Concentration of $50\,\mathrm{cm}^{-3}$ for an entrainment factor of $0.8$ and $0.6$, respectively.

Martin et al. (1994) showed that, for a measured cloud droplet size distribution, the effective and mean volume radius can be connected by an empirical and averaged factor $k$ ($k \approx 0.80 \pm 0.07$ for maritime clouds). Thus, it can be shown that the Cloud Droplet Number Concentration for maritime clouds is

$$N = \frac{3 LWC_{ad}}{4\pi\rho_w} \frac{1}{k} \left(\frac{100}{r_{\mathrm{eff}}}\right)^3 , \qquad (17)$$

5    where $\rho_w$ is the density of liquid water, $N$ is in units $\mathrm{cm}^{-3}$ and $r$ is in units µm (e.g. Reid et al., 1999). In case of entrainment we know that the measured $LWC$ will be different from the theoretical $LWC_{ad}$. Therefore, we must consider Equation 16. Thus, Equation 17 yields

$$N_{en} = \frac{3 \cdot a \cdot LWC_{ad}}{4\pi\rho_w} \frac{1}{0.8} \left(\frac{100}{r_{\mathrm{eff}}}\right)^3 . \qquad (18)$$

Moreover, we can reconstruct the effective radius profile using Equation 18. The effective radius profile becomes

10    $$r_{\mathrm{eff}}(z_c) = 100 \cdot \left(\frac{3 \cdot a \cdot LWC_{ad}(z_c)}{4\pi\rho_w N_{en}} \frac{1}{0.8}\right)^{1/3} , \qquad (19)$$

where $N_{en}$ is given in units $\mathrm{cm}^{-3}$ and the radius $r_{\mathrm{eff}}$ in units µm. In this case, we assume that the Cloud Droplet Number Concentration $N_{en}$ remains constant within the cloud. The right part of Figure 5 shows three effective radius profiles for three different $N$ with identical initial temperature and pressure values at cloud base. The dashed and dotted orange lines show the profiles for an entrainment factor $a$ of $0.8$ and $0.6$, respectively.





## 4   Spectral Re-Sampling of Active Remote Sensing Data

Optical thickness, effective radius and thermodynamic phase can be retrieved for the full specMACS field-of-view. Quantities such as CTH and LWP can only be determined for the nadir direction directly below the aircraft where lidar and radiometer measurements exist. One possible approach to spread the information of nadir measurements to the wider specMACS field-of-view is described by Barker et al. (2011).

Figure 6 shows the basic idea of Barker et al. (2011). The spectra between nadir columns and off-nadir columns are compared with each other within about $25\,\mathrm{km}$ distance to the observer. If the spectra match, the nadir-measured remote sensing information of another instrument is transferred to the off-nadir column. The nadir pixel will then be called *donor* and the off-nadir pixel will be called *recipient*.

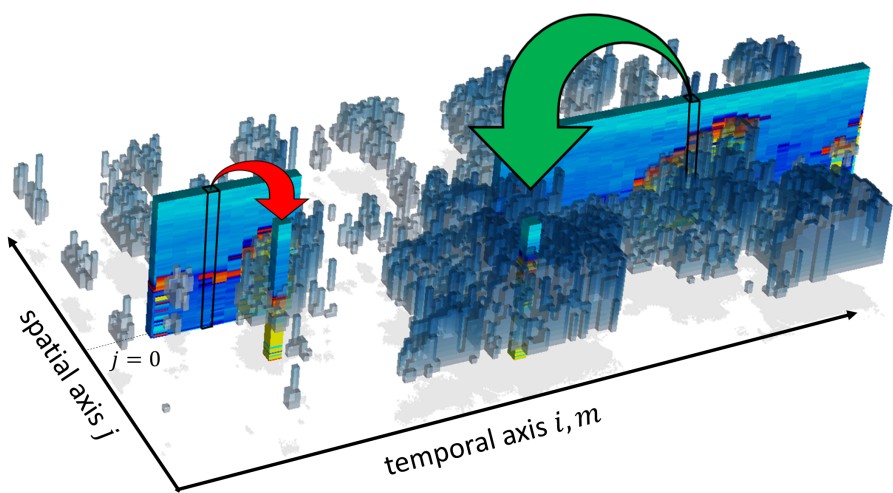

**Figure 6.** The main idea of Barker et al.: The information of the remote sensing instruments measuring only below the aircraft will be extended to the broader pushbroom specMACS field-of-view. The green arrow shows a good match and the red arrow a bad match between the two columns.

This mapping is possible since the spectrum contains information about cloud macro- and microphysics. For example, the spectrum at two different cloud locations with the same geometrical thicknesses but with two different LWPs will look different. Figure 7 shows the shortwave infrared spectrum of specMACS for different LWP values at a constant CTH (blue) and for different CTHs at constant Liquid Water Path values (dashed, red). We see, for example, between 1400 and $1800\,\mathrm{nm}$ that the radiance decreases with increasing LWP values because the cloud droplet effective radii usually become larger which means that the reflection becomes weaker in this wavelength range. Moreover, the radiance increases with increasing CTH





values in the same wavelength range, since the atmospheric absorption of sunlight by water vapor decreases with increasing CTH.

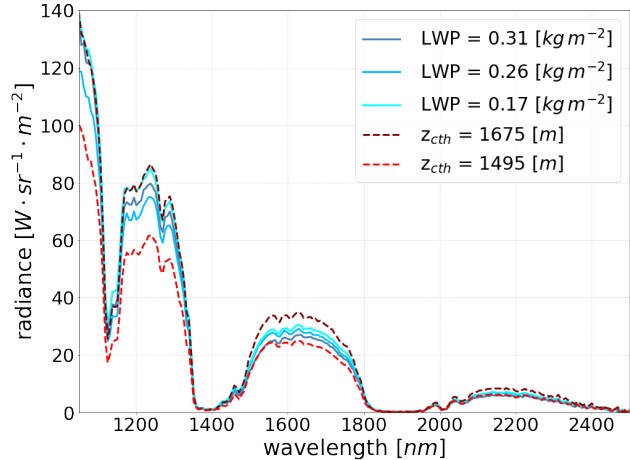

**Figure 7.** The measured shortwave infrared spectrum of specMACS for both different LWP values at a constant CTH (blue) and different CTH values at a constant Liquid Water Path (dashed, red).

The algorithm is described and verified by Barker et al., and in our case, applied for the shortwave infrared sensor of the specMACS operating between $1015$ and $2496\,\mathrm{nm}$. Similar to Barker et al., we can define the deviation between the donor spectrum and recipient spectrum using the Euclidean distance $d$

$$d_{ijm} = \sqrt{\sum_\lambda (L_{ij\lambda} - L_{m0\lambda})^2}, \tag{20}$$

where $i$ is the temporal axis and $j$ the spatial axis of the specMACS, $m$ is the temporal axis and $0$ the spatial location of the nadir pixel, $\lambda$ the wavelength and $\boldsymbol{L}$ the radiance. The donor pixel $D$ is the pixel with the smallest distance $d$:

$$D_{ij} = \arg\min_m \left( \sqrt{\sum_\lambda x_{ijm\lambda}^2} \right) \quad \text{with} \quad x_{ijm\lambda} \equiv L_{ij\lambda} - L_{m0\lambda} \tag{21}$$

The Euclidean distance alone does not include physics. In the shortwave infrared region, radiances at smaller wavelengths are higher than radiances at longer wavelengths because the solar radiation is more powerful. Subsequently, radiances at shorter wavelengths contribute more to the distance and radiances at longer wavelengths contribute less to the distance. Therefore, we introduce a weight $\boldsymbol{\omega}$ which we define as

$$\omega_\lambda = \frac{1}{L_{toa,\lambda}^2}, \tag{22}$$

where $\boldsymbol{L_{toa,\lambda}}$ is the radiance at the top of the atmosphere calculated by using Kurucz (1992) in libRadtran. This weight decreases the contribution to $d$ of radiances at shorter wavelengths and it increases the contribution at longer wavelengths.





Using the weight in Equation 21 yields:

$$D_{ij} = \arg\min_{m} \left( \sqrt{\sum_\lambda \omega_\lambda \cdot x_{ijm\lambda}^2} \right). \tag{23}$$

We got initial good results with this weight. We also, for example, increased the weight at radiances where water vapor absorption is enhanced, but did not see any significant influence on the result. We also tested the algorithm following the suggestion
5   of Barker et al. (2011) to reconstruct the nadir cross-section and got similar results (not shown).

The data amount is, with about $120\,\mathrm{GB}$ each flight, not small. In Figure 7 not only differences between the spectra are visible, but also similarities. Thus, it should be possible to describe most of the spectra with fewer parameters which can be done with Principal Component Analysis (PCA, see e.g. Jolliffe (2002)). We apply the PCA on the measured radiance $\boldsymbol{L}$

10   $$L_{ij\lambda} = \sum_c P_{\lambda c} \widetilde{L}_{ijc} + k, \tag{24}$$

where $\boldsymbol{P}$ is the PCA transformation, $\widetilde{\boldsymbol{L}}$ is the radiance in the PCA notation, $\boldsymbol{L}$ is the measured radiance, $c$ the PCA components and $k$ is the mean value of the measured radiance. Furthermore, we yield for $\boldsymbol{x}$:

$$\begin{aligned} x_{ijm\lambda} &= L_{ij\lambda} - L_{m0\lambda} \\ &= \sum_c P_{\lambda c} \widetilde{L}_{ijc} + k - \sum_c P_{\lambda c} \widetilde{L}_{m0c} - k \\ &= \sum_c P_{\lambda c} \left( \widetilde{L}_{ijc} - \widetilde{L}_{m0c} \right) \\ &= \sum_c P_{\lambda c} \widetilde{x}_{ijmc}. \end{aligned} \tag{25}$$

Inserting the result into Equation 23 yields:

$$\begin{aligned} D_{ij} &= \arg\min_{m} \left( \sqrt{\sum_\lambda \omega_\lambda \cdot \left( \sum_c P_{\lambda c} \widetilde{x}_{ijmc} \right) \left( \sum_{c'} P_{\lambda c'} \widetilde{x}_{ijmc'} \right)} \right) \\ &= \arg\min_{m} \left( \sqrt{\sum_{cc'} \widetilde{x}_{ijmc} \cdot \sum_\lambda \underbrace{P_{\lambda c} \omega_\lambda P_{\lambda c'}}_{\equiv \widetilde{\omega}_{cc'}} \cdot \widetilde{x}_{ijmc'}} \right) \\ &= \arg\min_{m} \left( \sqrt{\sum_{cc'} \widetilde{x}_{ijmc} \cdot \widetilde{\omega}_{cc'} \cdot \widetilde{x}_{ijmc'}} \right). \end{aligned} \tag{26}$$

18 PCA components explain more than $99.7\,\%$ of the variability of the original signal. We tested the algorithm with 50 components without any visible changes in the results. Due to the use of the PCA, we save around a factor of ten in memory.





Having found the according donors, the CTH of the WALES lidar, the reflectivity of the HAMP radar and the LWP of the HAMP radiometer are mapped on the wider specMACS field-of-view. Only donors having a maximum distance of about $25\,\mathrm{km}$ from the observer are used. Barker et al. (2011) found that the usual distance between the recipient and the donors are less than $30\,\mathrm{km}$ and mostly below $5\,\mathrm{km}$. We assume that meteorological conditions are mostly similar within that distance.

### 4.1 Macrophysics

Figure 8 shows the CBH derived with dropsonde measurements on the left and the CTH derived with the combination spec-MACS and WALES following the spectral re-sampling approach on the right. The CBH is constant for all clouds and is about $600\,\mathrm{m}$. Ocean areas are indicated by a Cloud Bottom Height of $0\,\mathrm{m}$. The CTH at cloudy pixels range from about 800 to $1700\,\mathrm{m}$.

10  Over the cloud-free ocean areas, the donor pixels are influenced by the sun reflection and cloud shadows. Therefore, we see over the ocean in some cases a signal.

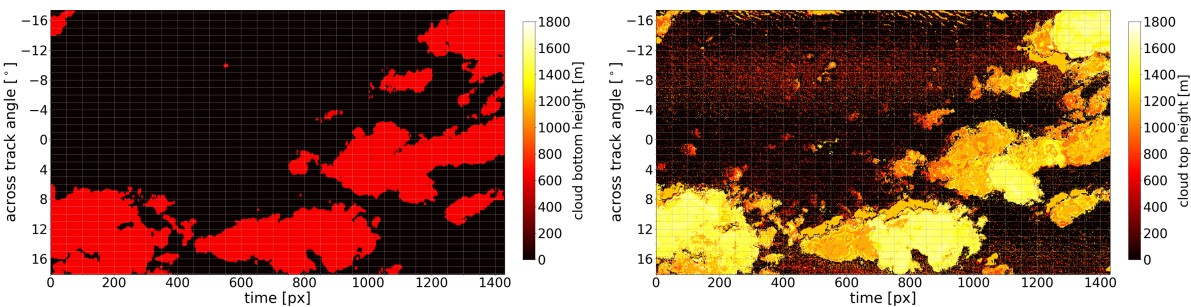

**Figure 8.** The left part shows the CBH with an applied cloudmask retrieved from the Lifted Condensation Level of a dropsonde at a height of about $600\,\mathrm{m}$. The right part shows the corresponding CTH where no cloudmask is applied.

Knowing the velocity $\boldsymbol{v}$, the height $z_{plane}$, as well as the roll, yaw and pitch angles of the aircraft, the height of the cloud $z_{cth}$ and the viewing angles of all camera pixels, we can transform the CTH points $\boldsymbol{p_b}$ seen from the specMACS SWIR camera into the three-dimensional reconstruction following four steps:

Firstly, we put the origin of the *local reference system* in the middle of one cloud scene (see also Figure 9). Secondly, we calculate the CTH points $\boldsymbol{p_b}$ and transform them from the specMACS SWIR camera coordinate system into the *local reference system* with $15\,\mathrm{m}$ box resolution following

$$\boldsymbol{p}_b = \boldsymbol{O}_{\mathrm{SWIR}} + \frac{\boldsymbol{v}_d \cdot (z_{plane} - z_{cth})}{\cos(\alpha)} \,. \tag{27}$$

20  $\alpha$ is the viewing angle of the SWIR camera pixel, $\boldsymbol{O}_{\mathrm{SWIR}}$ the origin of the specMACS SWIR camera which is in the focal point of the optics and $\boldsymbol{v}$ is the direction from $\boldsymbol{O}_{\mathrm{SWIR}}$ to the points $\boldsymbol{p_b}$





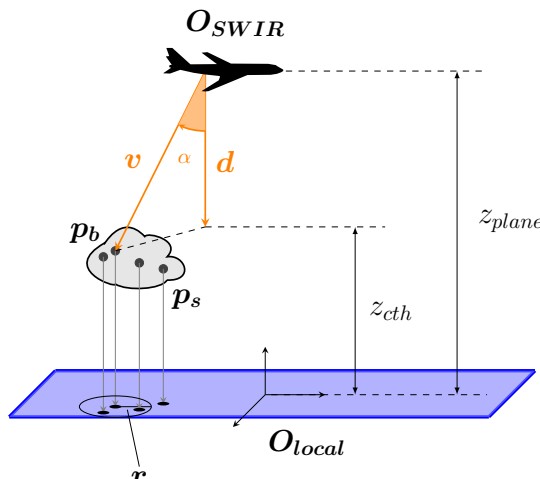

**Figure 9.** Schematic illustration of vectors used to transform the CTH points $p_b$ into the local reference system $O_{local}$. The points $p_s$ are all points in a circle around $p_b$ having a radius $r$.

Thirdly, we average all points $p_s$ within a circle of $5\,\mathrm{m}$ radius. Thereby, and for performance reasons, a K-D Tree (Bentley, 1975) was used to create a space-partitioning data structure to find for each CTH point $p_b$ all points $p_s$ within a circle of $5\,\mathrm{m}$ radius (circle on plain in Figure 9). Single gaps can occur when the cloud blocks the specMACS SWIR view to pixel lying behind. These gaps are filled with the average height of the surrounding pixel.

Finally, we use the theoretical determined CBH $z_{cbh}$ of the nearest dropsonde and fill the columns up to the WALES retrieved CTH $z_{cth}$ with microphysical information as described in the following.

## 4.2 Microphysics

Lookup tables are calculated for the optical thickness and the effective radius for different atmospheric parameters and solar
10    positions using a plane-parallel approach. The radiance at $750.2\,\mathrm{nm}$ is used to derive the optical thickness and the radiance at $2160\,\mathrm{nm}$ is used to derive the effective radius similar to Nakajima and King (1990). Comparing the measured values to the simulated values allows to determine the optical thickness and effective radius. The thermodynamic phase is calculated following Jäkel et al. (2013) as described before. The LWP was provided by Jacob et al. (2019a) from the HAMP radiometer and mapped on the specMACS field-of-view.

Figure 10 shows on the left part the effective radius and on the right part the corresponding optical thickness with an applied cloudmask. Both quantities have been derived with specMACS alone. The effective radius over the ocean is significantly





higher than at cloud areas and increases to values above $25\,\mu$m. Inside cloud areas, the effective radius has values between 6 and $17\,\mu$m. On cloud edges, as well as at cloud shadow areas, the effective radii values increase strongly. We also see a wave pattern inside the ocean caused by sun reflection. The optical thickness increases in cloud center to values of about 35. On cloud borders, the optical thicknesses is below 5.

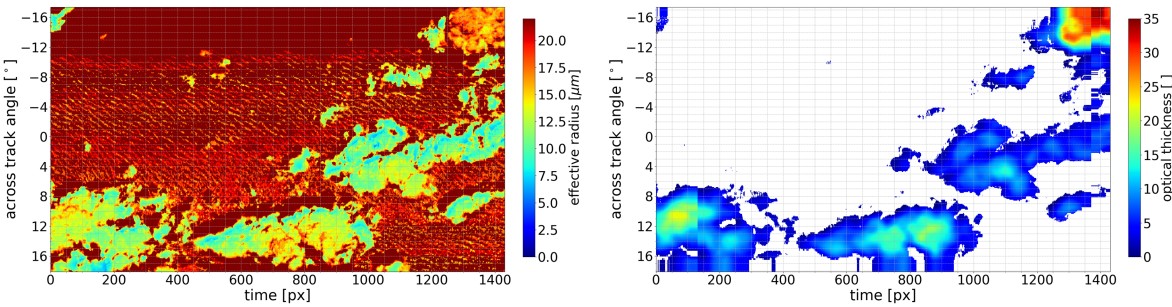

**Figure 10.** The left part shows the effective radius. The right part shows the optical thickness with applied cloudmask.

5     Figure 11 shows on the left the LWP derived with the HAMP radiometer and extrapolated to the specMACS field-of-view using the spectral re-sampling approach (see Section 4). On the right, the thermodynamic phase index with cloudmask derived by specMACS alone is shown. The LWP values increase in cloud centers to values of about $0.2\,\text{kgm}^{-2}$. The reflection in the ocean also creates a weak signal and the shadows of the clouds remove any signal. The thermodynamic phase index is negative which indicates liquid clouds. Considering the dropsonde profile (not shown), the temperature of the atmosphere was positive

10     to altitudes of about $4500\,\text{m}$. Thus, the clouds are pure liquid clouds.

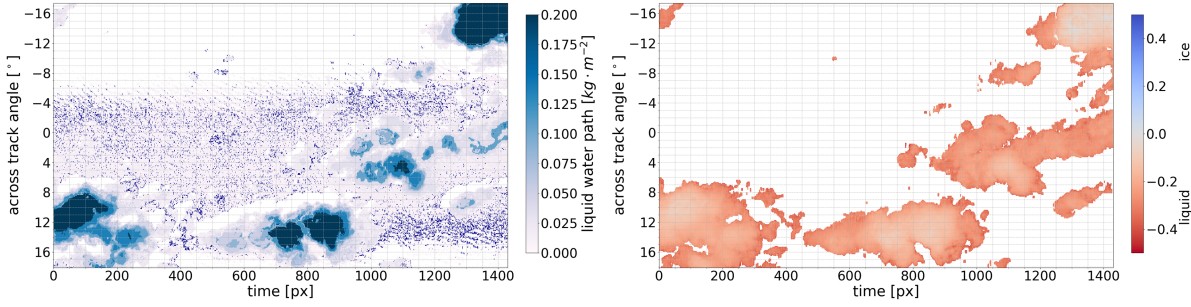

**Figure 11.** The left part shows the liquid water path. The right part shows the thermodynamic phase index with applied cloudmask.

The Liquid Water Content Profile is derived with the adiabatic equations (Equation 15) using dropsond data, the CBH, the CTH and the LWP. We adapt this profile with the factor $a$ so that the theoretical LWP matches the measured Liquid Water Path


from the HAMP radiometer according to Equation 16.

## 4.3 Filtering

The following filtering method is very strict and eliminates about $90\,\%$ of the available data but has to be done to find the pixels in a cloud domain where the effective radius retrieval of Nakajima and King (1990) works well with a high certainty. With the filtered effective radius values we will then retrieve the Cloud Droplet Number Concentration for this cloud domain.

The calculation of effective radius profiles and corresponding Cloud Droplet Number Concentrations is subject to uncertainties caused by cloud inhomogenities: Marshak et al. (2006) confirmed that shadowing in one-dimensional retrievals results in a large overestimation of effective radii and showed that effective radius retrievals for thin clouds have a high uncertainty (Zinner et al. (2010); Marshak et al. (2006)). Ice particles absorb shortwave infrared radiation differently than water (Hansen and Travis, 1974). Besides photon losses at cloud edges, illumination effects and horizontal photon transport, which is not considered in one-dimensional retrievals, influence the result (Stevens et al. (2019); Zeng et al. (2014); Zhang and Platnick (2011); Zinner et al. (2010)). Hence, we can only retrieve the effective radius for shadow-free areas, optical thick clouds, ice free areas and cloud parts with less three-dimensional effects.

We made the assumption that the effective radius usually grows with height (see Subsection 3.3). When the retrieved effective radii vary within the same cloud height significantly, the values are probably affected by shadows, three-dimensional photon losses or gains, ice effects or, in the worst case, by entrainment (the mixing of ambient clear sky air into the cloud). One way to define the quality of the retrieved effective radii for the whole effective radius profile is the standard deviation averaged over all contributing cloud layers which we define as

$$\overline{\sigma} = \sqrt{\frac{1}{M} \cdot \sum_{j}^{M} \left( \frac{1}{N} \cdot \sum_{i}^{N} \left( \bar{r}_{\mathrm{eff}\,j} - r_{\mathrm{eff}\,ij} \right)^2 \right)}, \tag{28}$$

where $i$ is the number of effective radii each layer, $j$ the number of discrete height layers, $r_{\mathrm{eff}}$ the effective radii and $\bar{r}_{\mathrm{eff}}$ the mean effective radii of all effective radii $r_{\mathrm{reff}\,i}$ at height layer $j$. In an idealized case and without mixing effects this mean standard deviation would be very close to zero.

First, we apply the cloudmask as described in Appendix B. Then, we filter areas that contain shadows using ray tracing. Figure 12 shows an example of the shadow mask. The area of a ray will be identified as shadow if it intersects with another cloud surface or when it is close to one. In addition, we exclude all ice parts of the cloud. We then calculate the mean standard deviation for increasing optical thickness thresholds and for increasing geometrical distance thresholds from the shadow areas.





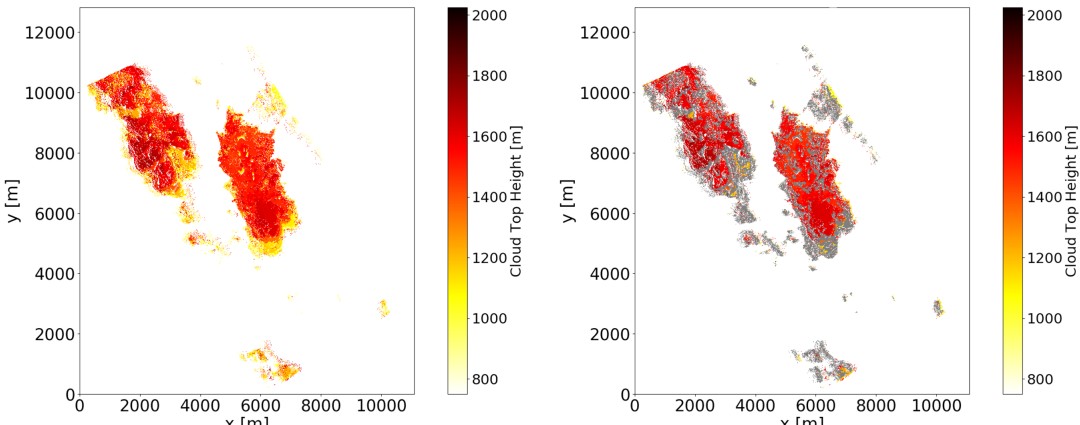

**Figure 12.** An example of an applied shadow mask. The left figure shows the CTH of a three-dimensional reconstructed scene from above without the shadow mask. The right figure shows the same scene with the shadows in gray. The $y$ axis is directed southwards and $x$ axis is directed along westwards. The solar zenith angle is about $3.6°$ and the azimuth is south-east ($140°$). Thus, the sun shines from the upper left corner of the figure.

After an applied optical thickness threshold of above roughly 8, the mean effective standard deviation does not decrease so much anymore. Thus, we removed all parts of the clouds having a smaller optical thickness than 8. Marshak et al. (2006) also showed in their Figure 5 that optical thicknesses smaller than 10 are connected to strong fluctuations. Thus, an optical thickness threshold of 8 was applied and the mean standard deviation could be reduced from about 4.8 to below $3\,\mu m$.

Moreover, we found that the mean effective standard deviation does not decrease strongly for more than $45\,m$ geometrical distance from shadow areas. Thus, a geometrical distance threshold of $45\,m$ was applied. The mean standard deviation in this case could be reduced from about 4.8 to about $2.5\,\mu m$. Combining both filters, the mean standard deviation could be reduced in most cases to values below about $2\,\mu m$.

Applying these thresholds on our data, we hypothesize that the areas left over are mostly not influenced by three-dimensional effects, shadow, ice, or surface reflectivity. Figure 13 shows the retrieved effective radius profile and the theoretical one. The increase of the effective radius with cloud height (see also Figure 13) is only visible after this described filtering method.





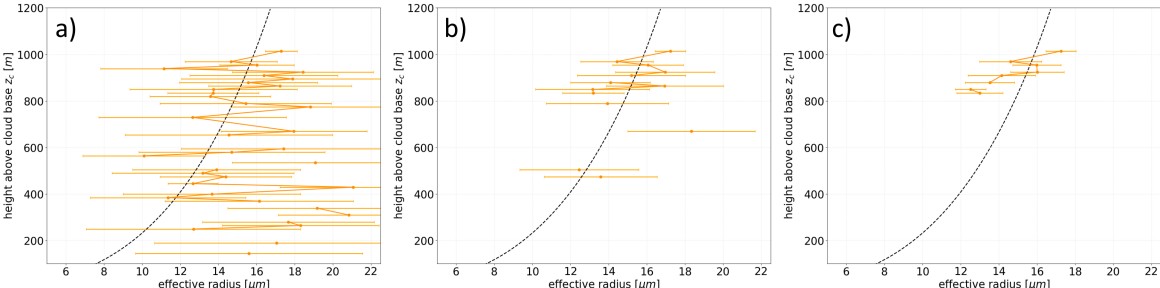

**Figure 13.** Retrieved effective radius profile (orange) and a theoretical expected profile (black dashed) for a cloud scene with about $31\,\%$ cloud cover measured on 19th August 2016. The figure on the left shows the retrieved effective radius for the heights above cloud base without any filtering applied. The figure in the middle shows the effective radius for an optical thickness threshold of 8 and a geometrical distance threshold of $15\,\mathrm{m}$ from shadow areas. The figure on the right shows the effective radius with the full filtering applied, i.e., optical thickness threshold of 8 is applied and geometrical distances of $45\,\mathrm{m}$ from shadow areas are removed.

Figure 14 presents a two-dimensional illustration of the filtering method. It shows the unfiltered effective radius in the Cartesian local reference coordinate system on the left and on the right the filtered effective radius. North is directly upwards $y[m]$ and east is towards the right side $x[m]$. We see that only for the largest clouds and central areas some effective radius values are available after the filtering method. Strong effective radius fluctuations cannot be seen in these areas.

For these filtered effective radius areas, the Cloud Droplet Number Concentration $N$ is calculated and averaged. It is $38 \pm 15\,\mathrm{cm}^{-3}$ in this cloud scene. Moreover, the best theoretical effective radius is fitted onto the retrieved and filtered effective radius data and used for all clouds within this cloud scene.





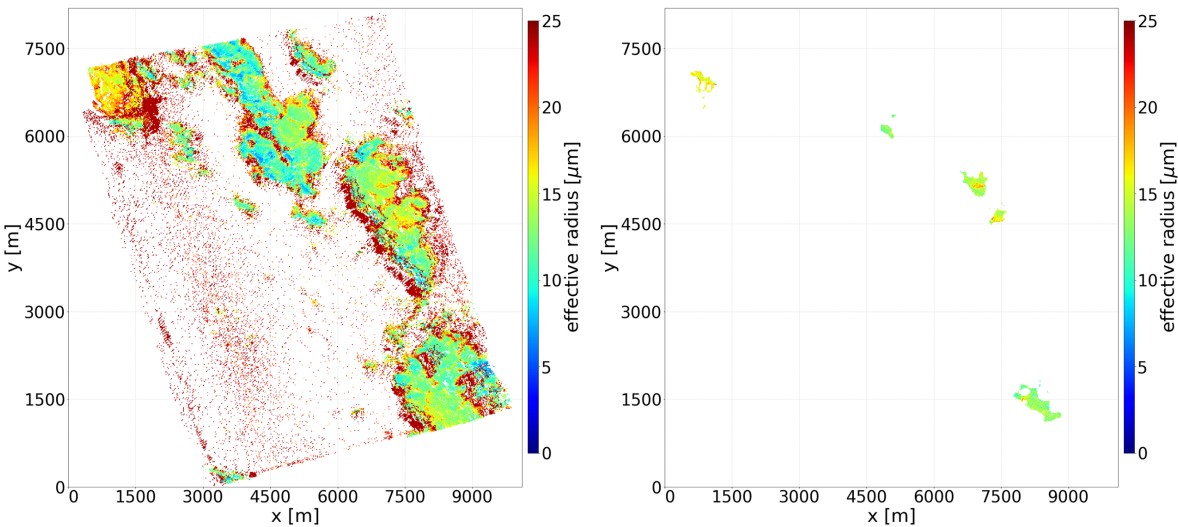

**Figure 14.** The left part shows the unfiltered effective radius. The right part shows the filtered effective radius. The $y$ axis is directed northwards and the $x$ axis is directed eastwards.

## 4.4 Three-Dimensional Reconstruction

A reconstructed cloud scene with about $35\%$ cloud cover is shown in Figure 15. This cloud scene shows reconstructed cumulus clouds different sizes and was taken on the HALO flight on 19th August 2016. It is about $6\,\mathrm{km}$ wide and $12\,\mathrm{km}$ long. The highest cloud geometrical thickness is about $1000\,\mathrm{m}$ and the smallest cloud is about $200\,\mathrm{m}$ thick. The maximum CTH value

is about $1700\,\mathrm{m}$ and the cloud bottom height is constant at $625\,\mathrm{m}$. The two-dimensional surface area of a cloud viewed from above varies between about $0.1$ and $8\,\mathrm{km}^2$.

The upper part shows the three-dimensional effective radius and the lower part shows the liquid water content. The effective radius increases very strongly in the first one hundred meters roughly and peaks in this case at about $14\,\mu\mathrm{m}$. The Liquid Water

Content increases with height and peaks at about $0.3\,\mathrm{g\,m}^{-3}$. The retrieved Cloud Droplet Number Concentration and the corresponding standard deviation for the whole scene is $38\pm26\,\mathrm{cm}^{-3}$. In most other cases the standard deviation is between $5$ and $15\,\mathrm{cm}^{-3}$.

We see single spikes on the cloud edges with high Cloud Geometrical Thicknesses. We see also that the effective radius is

identical in every cloud. Entrainment effects and mixing processes, especially on the cloud borders, are, in the effective radius profile, not considered. Also the constant CBH is visible. The Liquid Water Content is not identical in every cloud since it is derived via the LWP of the HAMP.



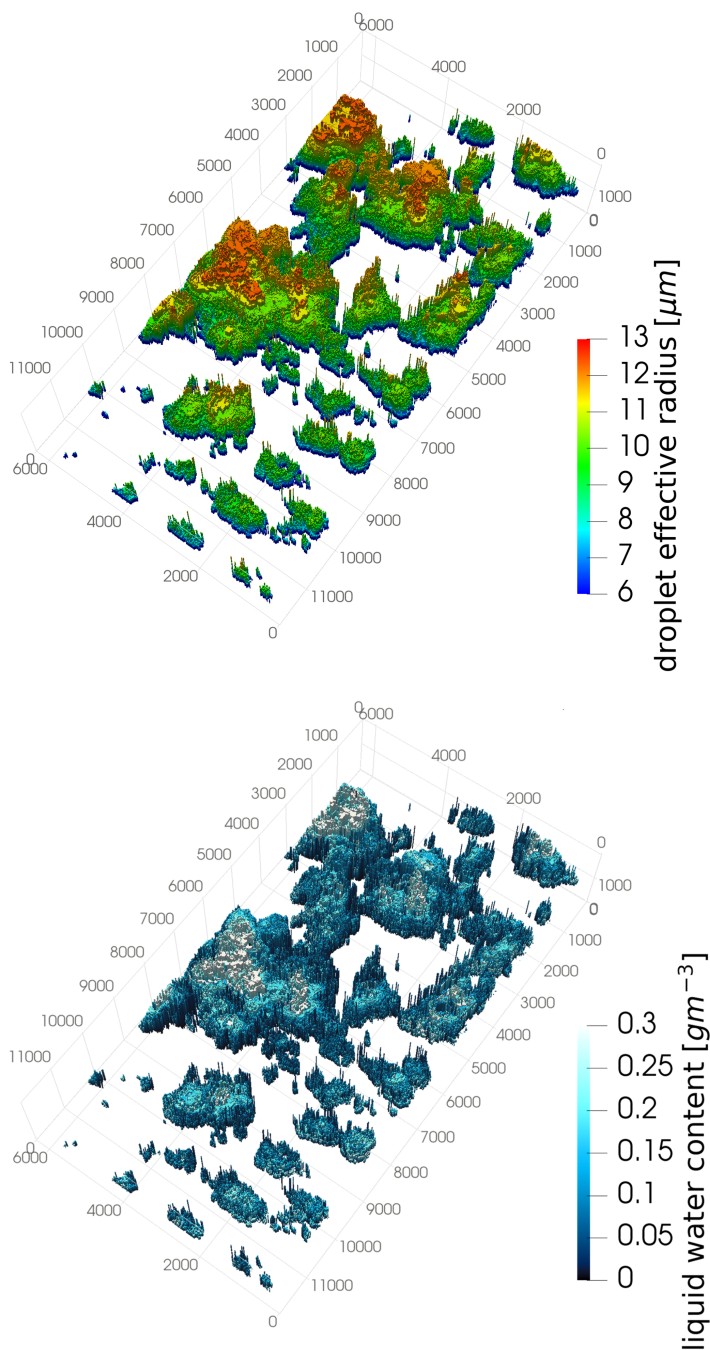

**Figure 15.** Reconstructed cloud macrophysics including the three-dimensional effective radius (upper) and liquid water content (lower) distribution. The axis are in units m.



## 4.5 Consistency Check

The quality of the reconstructed three-dimensional cloud fields will be examined in the following. Since in-situ measurements for a validation against real data are not available, theoretical experiments with simulations are performed. Thereby, the radiative transfer software MYSTIC included in libRadtran is used (see Section 2.7). Based on reconstructions introduced above, the radiance field observable by specMACS is simulated. The resulting synthetic measurement is then compared to the true measurement. Thus, general deficiences as well as limitations of our method should become visible. We use simulations at two wavelengths, 750.2 and 1553.5 nm. 750.2 nm is a wavelength in the visible spectrum and is mainly sensitive to the optical thickness. 1553.5 nm is a wavelength in the shortwave infrared spectrum and has additional sensitivity to the effective radius. The simulated radiance, the measured radiance and the corresponding histograms for 750.2 nm are shown in the left part of Figure 16. The right part, shows the same for the radiance at 1553.5 nm.

The two histograms in Figure 16 show the reflectivity distribution for only the cloudy parts. In order to further facilitate the assessment of the reconstruction quality, additional tests are conducted. The dashed lines in the histogram at 750.2 nm (c) show simulation results for cloud data sets with 20 % increased (dashed blue line) and decreased (dashed cyan line) liquid water path values, equivalent to a $\pm 20\,\%$ cloud optical thickness change. The dashed lines in the histogram for 1553.5 nm (f) show simulations with 20 % increased (dashed blue line) and decreased (dashed cyan line) effective radius at constant optical thickness.

The shadows on the sea surface at both wavelengths show that CTHs and CBHs are reconstructed well. However, the extent of the shadows is larger for the simulation and more pronounced. This is mainly a consequence of missing variability of the CBH measurement which is replaced by a constant Convective Condensation Level representing the lowermost possible cloud height.

At 750.2 nm, the reflectivity is sensitive to the optical thickness. In general, the simulated radiances (b) at the cloud centers of the smaller clouds agree well with the measured radiances (a). However, in the cloud centers of the two biggest clouds close to the edge of the specMACS field-of-view, differences of up to roughly $65\,\mathrm{W \cdot sr^{-1} \cdot m^{-2}}$ can be seen. Furthermore, cloud edges appear brighter in the simulation. The averaged reflected radiance of the cloudy areas is $109.86\,\mathrm{W \cdot sr^{-1} \cdot m^{-2}}$ in the simulation and $98.57\,\mathrm{W \cdot sr^{-1} \cdot m^{-2}}$ in the measurement. This is a difference of about $11.4\,\%$ with higher values in the simulation. The radiance distributions at 750.2 nm are shown in the histograms in Figure 16c.

At 1553 nm, the forward simulation shows additional sensitivity to liquid water absorption, and thus, to droplet effective radius. Looking into details we find additional differences compared to the comparison at 750.2 nm. For example, the level of brightness in the simulation is, in general, elevated which is especially obvious in cloud centers. Moreover, cloudy areas appear larger because cloud edges are brighter. The averaged reflected radiance of cloudy areas is $20.91\,\mathrm{W \cdot sr^{-1} \cdot m^{-2}}$ in

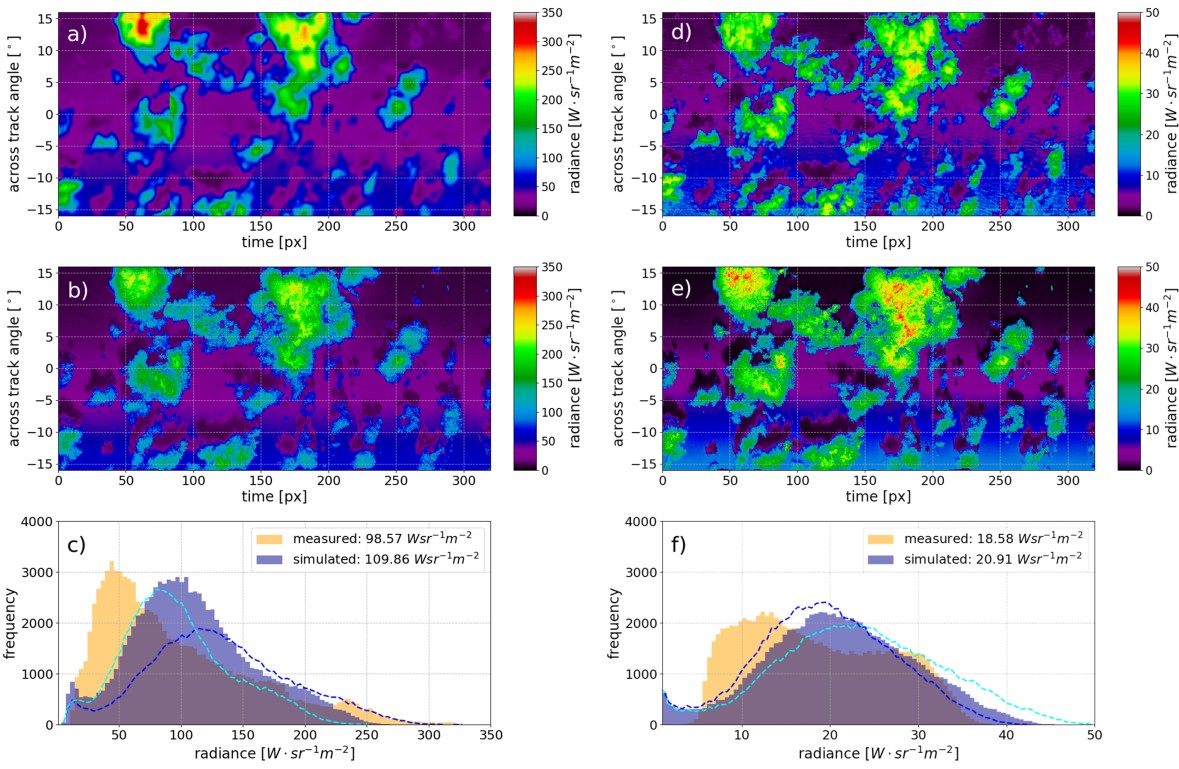

**Figure 16.** Consistency check for the three-dimensional reconstructed cloud scene. (a) shows the measured radiance at $750.2\,\mathrm{nm}$, (b) shows the simulated radiance at $750.2\,\mathrm{nm}$ and (c) shows the frequency of the simulation radiance values (blue) and measurement values (orange) at $750.2\,\mathrm{nm}$. Moreover, the dashed blue line shows the distribution for $20\,\%$ larger and the dashed cyan line shows the distribution for $20\,\%$ smaller LWP values. (d) shows the measured radiance at $1553.5\,\mathrm{nm}$, (e) shows the simulated radiance $1553.5\,\mathrm{nm}$ and (f) shows the frequency of the simulation radiance values (blue) and measurement values (orange) at $1553.5\,\mathrm{nm}$. Moreover, the dashed blue line shows the distribution for $20\,\%$ larger effective radius values and the dashed cyan line shows the distribution for $20\,\%$ smaller effective radius values. The LWP is scaled so that the cloud optical thickness remains constant.

the simulation and $18.58\,\mathrm{W}\cdot\mathrm{sr}^{-1}\cdot\mathrm{m}^{-2}$ in the measurement. This is a difference of about $12.5\,\%$ with higher values in the simulation.

The differences of the radiances at $750.2\,\mathrm{nm}$ between measurement and simulation has to be investigated further. At $750.2\,\mathrm{nm}$ the radiance distribution of the simulation in Figure 16c shows a higher occurrence of median radiance values (at about $80\,\mathrm{W}\cdot\mathrm{sr}^{-1}\cdot\mathrm{m}^{-2}$ to $150\,\mathrm{W}\cdot\mathrm{sr}^{-1}\cdot\mathrm{m}^{-2}$). In addition, the simulated peak values are smaller than the measured ones. Both differences are possibly caused by the rather wide $5°$ field-of-view of the HAMP radiometer. The wide field-of-view tends to average out large LWP values to lower values which will result in a lower reflectivity and small LWP values become larger

which result in a higher reflectivity. This is shown in Figure 17. (a) shows the measured radiance at 750.2 nm, (b) shows the measured radiance averaged to the HAMP resolution, (c) shows the simulated radiance and (d) shows the value distribution. We see that the largest measured radiance values become smaller and the smallest measured radiance values become larger when we reduce the measured radiance to the HAMP resolution. Also the peak simulated radiances agree well with the measured

5    radiances reduced to the HAMP footprint.

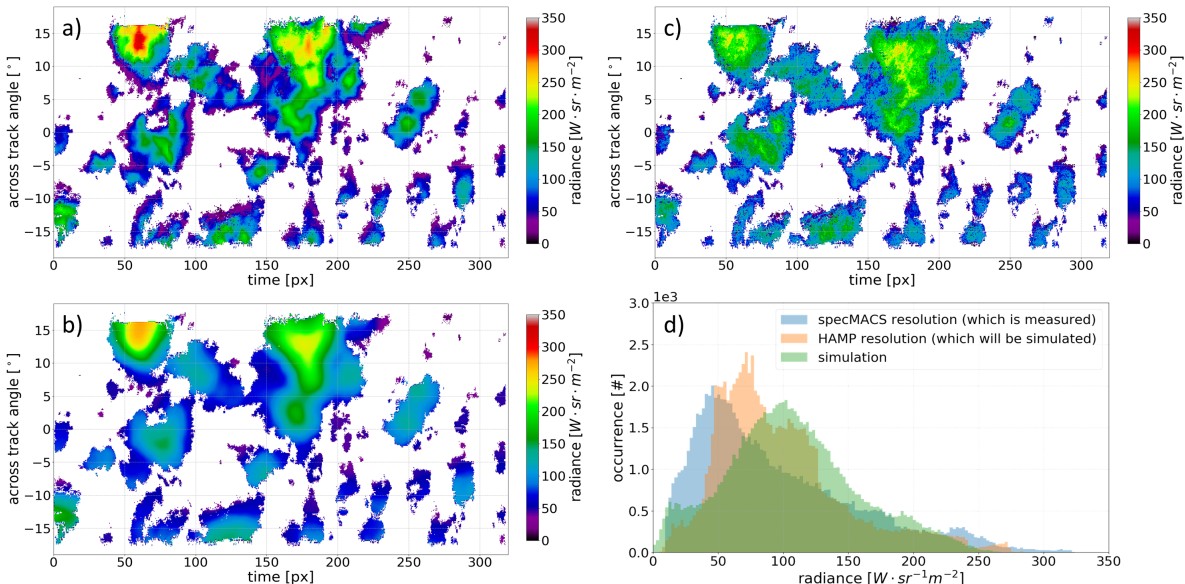

**Figure 17.** Effect of reducing the measurements to the HAMP resolution. (a) shows the measured specMACS radiance at 750.2 nm, (b) shows the measured specMACS radiance at 750.2 nm reduced to the HAMP resolution and (c) shows the simulated radiance values. The blue color in the distribution (d) represents the measured radiance at 750.2 nm, the orange color represents the measurements reduced to the HAMP resolution and the green color is the distribution of the simulated values of the reconstructed cloud.

The resolution problem explains most of the differences but not all. Further errors at 750.2 nm might be caused by a lack of matching donor pixels providing the necessary large LWP value. Figure 18 shows the deviation $d_{ijm}$ as introduced in Section 4 which also could be used as a quality filter for the matches. Especially at the center areas of the biggest clouds the deviation is high ($d_{ijm} > 0.010\,\mathrm{W\,sr^{-1}\,m^{-2}}$) indicating probably erroneous donor-recipient matches. For the rest of the cloud, the deviation

10    is small ($d_{ijm} < 0.010\,\mathrm{W\,sr^{-1}\,m^{-2}}$) indicating fitting donor-recipient matches.

The differences of the radiances at 1553.5 nm between measurement and simulation has to be inspected further. Source of the additional brightness at 1553.5 nm could either be a general overestimation of optical thickness or an underestimation of effective radius by our reconstruction method. An overestimation of optical thickness is not likely because this would mean





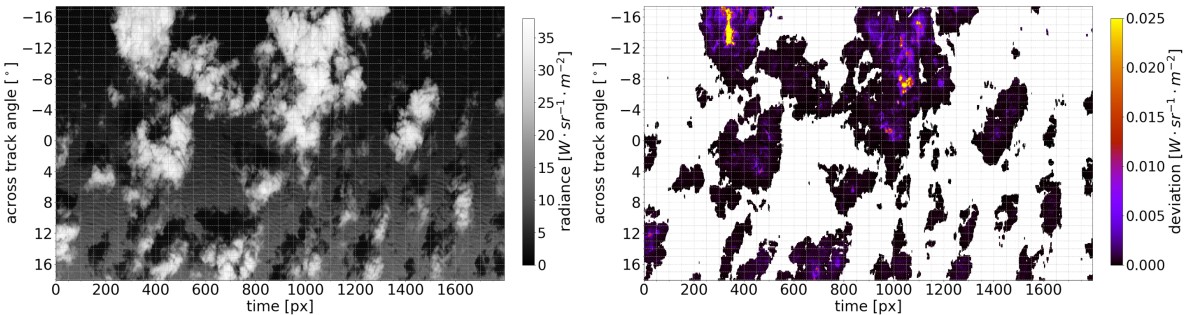

**Figure 18.** The radiance at $1553\,\mathrm{nm}$ is shown on the left and the deviation $d_{ijm}$ between donor spectrum and recipient spectrum is shown on the right. High deviation values are represented by yellow and red colors.

that the brightness of the simulated radiance values at the visible wavelength must be larger than the measured radiance values. This is not the case. An underestimation of the reconstructed effective radius would be possible. While shadow affected cloud areas and cloud edges are removed by the filtering method (tendency to strongly overestimate effective radius), bright illuminated cloud parts remain in our data (tendency to underestimate effective radius). These three-dimensional illumination effects
would lead to smaller effective radii (Zhang and Platnick (2011), Marshak et al. (2006)).

In order to put the quality of our reconstruction with its remaining limitations into perspective, the additional simulations can be investigated (dashed lines in Figure 16). The simulated base case shows signs of an underestimation of effective radius because the simulated radiance values are higher than the measured radiance values. Consequently, the test with larger effective
radii (dashed blue line in Figure 16f) should provide a better match. $20\,\%$ larger effective radii decreases the difference between the measured and simulated base case from $12.5\,\%$ to $4.1\,\%$ with higher values in the simulation. A reduction of cloud effective radius by $20\,\%$ (dashed cyan line) causes a stronger overestimation in the simulation with a difference of about $19\,\%$ to the measurement. Table 1 concludes the additional investigations for $\pm5°$ and $\pm10°$ limited specMACS field-of-views. Thus, we conclude that the effective radius uncertainty is close to roughly $20\,\%$.

Evaluation of the $750.2\,\mathrm{nm}$ tests allows additional conclusions regarding the effective radius. The increase of LWP by $20\,\%$ at $750.2\,\mathrm{nm}$ (dashed blue line) leads to a better match of the brightest values but only at the cost of a general shift towards larger values over the full radiance range. The average deviation between the simulated base case and the measured cloudy radiance is then at around $17.8\,\%$ with higher values in the simulation. A LWP reduction by $20\,\%$ (dashed cyan line) leads to
a higher concentration of values around $80\,\mathrm{W}\cdot\mathrm{sr}^{-1}\cdot\mathrm{m}^{-2}$ resulting in a difference of $0.2\,\%$ between the simulated base case and the measurement with higher values in the measurement. If only the part of the data set is analysed which is closer to the central line of data beneath the aircraft (the "donor line" for LWP and CTH information) the picture changes slightly. Table 1





presents the additional investigations for $\pm 5°$ and $\pm 10°$ limited specMACS field-of-views. Thus, LWP and optical thickness uncertainty is most likely within the range of 20 %. This accuracy is not larger than the general accuracy of the microwave radiometer derived underlying LWP values (Jacob et al., 2019a).

**Table 1.** Summary of simulations at $750.2\,\mathrm{nm}$ (upper) and at $1553.5\,\mathrm{nm}$ (lower) for limited specMACS field-of-view areas. The effective radii and LWP are varied by 20 %. Shown are averaged radiance values in $\mathrm{W\,sr^{-1}\,m^{-2}}$.

| field-of-view | measured | simulated base case | $LWP \cdot 1.2$ | $LWP \cdot 0.8$ | $r_{eff} \cdot 1.2$ | $r_{eff} \cdot 0.8$ |
|---|---|---|---|---|---|---|
| all | 98.57 | 109.86 | 119.90 | 98.35 | 96.80 | 127.38 |
| $\pm 10°$ | 86.03 | 99.89 | 109.90 | 88.54 | 87.19 | 117.25 |
| $\pm 5°$ | 85.68 | 99.71 | 109.49 | 88.68 | 87.42 | 116.54 |
| field-of-view | measured | simulated base case | $LWP \cdot 1.2$ | $LWP \cdot 0.8$ | $r_{eff} \cdot 1.2$ | $r_{eff} \cdot 0.8$ |
| all | 18.58 | 20.91 | 22.05 | 19.06 | 19.37 | 23.02 |
| $\pm 10°$ | 17.16 | 19.31 | 20.57 | 17.38 | 17.88 | 21.33 |
| $\pm 5°$ | 17.22 | 18.93 | 20.09 | 17.09 | 17.54 | 20.82 |

## 5 Discussion and Conclusion

We presented an approach to combine different active and passive remote sensing instruments on-board the HALO research flights during the NARVAL-II campaign. This approach makes it possible to reconstruct, with a box-size resolution of $15\,\mathrm{m}$, three-dimensional cloud macro- and microphysics of liquid trade-wind cumuli.

The three-dimensional extent of clouds is provided by a combination of Cloud Top Height from WALES lidar measurements, Cloud Bottom Height from dropsonde air-parcel analysis and horizontal cloud mask from spectral imagery. Lidar information from the track below the aircraft is spread on the wider imager swath following the spectral re-sampling approach. In the same way, Liquid Water Path information provided by the microwave radiometer HAMP is also spread on the specMACS swath. This is the starting point for a reconstruction of three-dimensional cloud microphysics.

Cloud optical thickness and effective radius retrieved via passive spectral remote sensing are added to provide a consistent microphysical representation of the cloud. Microphysical profiles within the clouds are determined using the HAMP measurements and the retrieved effective radius as constraints for a microphysical sub-adiabatic model. A strict filtering method eliminates effective radius values which are most likely affected by three-dimensional radiative transfer effects. Both shadow areas within clouds and cloud areas having small optical thicknesses close to cloud edges are removed from further data analysis as they are prone to retrieval errors (as also noted by Marshak et al. (2006)). Based on the remaining values, a most likely



averaged effective radius profile and an averaged cloud droplet number concentration can be derived for each cloud domain.

In-situ cloud data were not available for the discussed NARVAL-II campaign cases. Therefore, we performed a consistency check using simulations. From these we estimate that the accuracy of liquid water and effective radius values of our cloud re-
construction is in the range of about $20\,\%$ of the absolute values. The LWP errors are most likely random and scene dependent. The effective radius shows a bias towards smaller values. This bias visible in the near-infrared wavelength tests can probably be explained by remaining three-dimensional illumination effects not considered in the plane-parallel effective radius retrieval (also compare, e.g., Zhang and Platnick, 2011).

The cloud droplet number concentration of randomly selected cloud scenes on 19th August corresponding to a flight path of about $100\,\mathrm{km}$ over the Atlantic Ocean vary between $27 \pm 7$ and $47 \pm 16\,\mathrm{cm}^{-3}$ using the standard deviation as error. Assuming an error of $10\,\%$ in the effective radius, the cloud droplet numbers are about $27 \pm 11$ and $47 \pm 21\,\mathrm{cm}^{-3}$. These values are consistent to the annual average close to Barbados as shown in Figure 1 by Roelofs et al. (2006) or to previous studies such as Brenguier et al. (2000). They are also in the range of similar cloud scenes retrieved by Wolf et al. (2019) using non-imaging
solar reflectivity observations. Nonetheless, we are confident that our strict filtering leads to a strong reduction of uncertainty. Using the horizontal coverage of our specMACS data, especially information on cloud horizontal extent and cloud surface orientation, we are able to reduce the most important impact of tree-dimensional radiation transport.

It has to be kept in mind that we use a simple sub-adiabatic microphysical model assuming that both the cloud droplet
number concentration and the effective radius profile remain constant in every cloud part. Entrainment effects which might explain detailed differences are not considered. Visible differences at cloud edges might be caused by using only one constant Cloud Bottom Height for the whole cloud domain. In reality, the Cloud Bottom Height likely rises towards the cloud edge due to mixing. Unfortunately, available cloud radar data was not suited to determine a more realistic cloud bottom height due to the limited sensitivity to the small droplets. A combination of this retrieval with stereoscopic retrieved cloud top structure,
as recently presented by Kölling et al. (2019) using specMACS, could lead to better results. A more detailed and more direct reconstruction of the upper cloud border at cloud edges would become possible.

In subsequent work the radiative diabatic contributions to the local energy budgets in these reconstructed cloud fields will be analysed as well as their influence on potential vorticity and, thus, on dynamics. This method described in this paper will
also be useful for the EarthCARE satellite mission since similar instruments will be used (Illingworth et al., 2015) as well as the upcoming NARVAL-II follow-up EUREC4A campaign (Bony et al., 2017).



*Data availability.* The specMACS data are available at https://macsserver.physik.uni-muenchen.de (accessed at: 18th November 2019) after requesting a personal account. The radar data of the NARVAL-II flight on 12th August 2016 used in this paper is described by Konow et al. (2019) and the dataset was published by Konow et al. (2018). The Liquid Water Path data of the NARVAL-II flight on 19th August 2016 is described by Jacob et al. (2019a) and the dataset was published by Jacob et al. (2019b). WALES and dropsonde data are made available

through the DLR Institute of Atmospheric Physics in the HALO database (German Aerospace Center, 2016, available at: https://halo-db.pa.op.dlr.de, last access: 13 August 2019).

## Appendix A:  Adiabatic Theory - Standard Equations

**(a)** The decrease of saturation pressure with height $e_{sat}(T_c(z_c))$ can be calculated using an approximated Clausius-Clapeyron Equation which is valid for temperatures $-35\,°C < T_c < 35\,°C$ and accurate up to $0.3\,\%$ in this range (Bolton, 1980):

$$e_{sat}(T_c(z_c)) = 611.2\,hPa \cdot \exp\left(\frac{17.67 \cdot T_c(z_c)}{T_c(z_c) + 243.5}\right) \tag{A1}$$

**(b)** The decrease of temperature inside the cloud $T_c(z_c)$ can be calculated using the moist adiabatic lapse rate $\Gamma_f$ which is defined as (e.g., Vallis, 2019, p.234)

$$\Gamma_f = \frac{g \cdot (1 + (L \cdot r)/(R_{dry} \cdot T(z_c)))}{c_p + (L^2 r)/(R_v T(z_c)^2)}, \tag{A2}$$

with the Mixing Ratio of liquid water $r$, the specific heat $c_p$, the latent heat of vaporization $L$, the specific gas constant for dry air $R_{dry}$ and the specific gas constant for moist air $R_v$. The Mixing Ratio of liquid water can be approximated by $r \approx 0.622 \cdot e_{sat}(T(z_{cbh}))/(p_{air}(z_{cbh}))$ because $e_{sat} \ll p_{air}$. Moreover, we approximate $T_c(z_c)$ with $T(z_{cbh})$ for shallow cumulus clouds since the vertical extent, as found in this study, is mostly less than $1000\,m$. Thus, the temperature differences inside the

clouds are usually less than $5\,K$. Thus, we can express the decrease of temperature inside the cloud with height as:

$$T_c(z_c) = T(z_{cbh}) - \Gamma_f \cdot z_c \approx T(z_{cbh}) - \frac{g \cdot (1 + (L \cdot r)/(R_{dry} \cdot T(z_{cbh})))}{c_p + (L^2 r)/(R_v T(z_{cbh})^2)} \cdot z_c \tag{A3}$$

**(c)** The decrease of the air pressure with height $p_{air}(T_c(z_c))$ can be calculated using the Barometric formula with $M_{dry}$ as the molar mass of the Earth's air, $g$ the gravitational acceleration and $R_0$ the universal gas constant:

$$p_{air}(T_c(z_c)) = p_{air}(z_{cbh}) \cdot \exp\left(-\frac{M_{dry} \cdot g}{R_0 \cdot T_c(z_c)} \cdot z_c\right) \tag{A4}$$

The pressure $p_{air}(z_{cbh})$ at the cloud base is used.


## Appendix B: Cloudmask

A simple cloudmask using brightness thresholds is in our case not possible because the reflected sun in the ocean, called sunglint, is often brighter than surrounding small clouds. Therefore, this cloudmask is based on the ideas of Gao and Kaufman (1995) who used absorption of water vapor at $1375\,\mathrm{nm}$ to detect cirrus clouds. In an atmosphere without clouds, solar radiation

5    penetrates to the low and moist regions of the troposphere before it is reflected by the Earth's surface. In an atmosphere with clouds, solar radiation will be reflected by the clouds at higher altitudes, and thus, both the passed distance and the water vapor absorption will be smaller. Figure B1 shows examples of the simulated spectra of the SWIR camera.

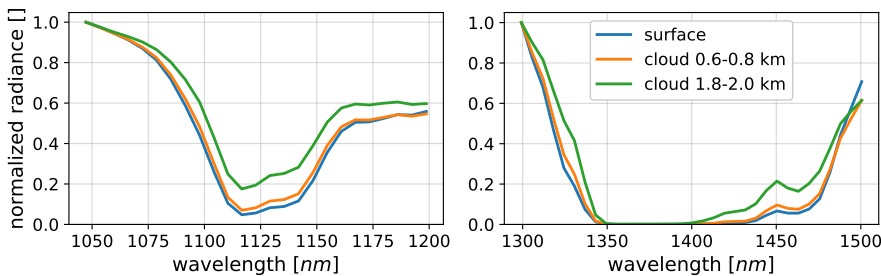

**Figure B1.** Simulated spectra of the SWIR camera using the DISORT solver included in libRadtran. The figure shows the water vapor absorption bands at 1125 (left) and $1375\,\mathrm{nm}$ (right) for three cases: the surface without clouds (blue), a cloud between 0.6 and $0.8\,\mathrm{km}$ and a cloud between 1.8 and $2.0\,\mathrm{km}$. The spectra are normalized to their respective maximum.

The cloud mask was generated according to the following steps: First, one reference spectrum with atmospheric molecular

10   absorption and one reference spectrum without atmospheric molecular absorption are simulated using the DISORT solver included in libRadtran. These spectra are then convolved with the spectral response function of the specMACS SWIR camera. Second, the transmittance $T_{ref,\lambda}$ of the reference atmosphere is determined by dividing the simulated spectrum with absorption by the simulated spectrum without absorption. Finally, the product of the spectrum without absorption but with transmittance is fitted to each measurement following

$$L_{meas,\lambda} = a \cdot L_{noabs,\lambda} \cdot (T_{ref,\lambda})^x \,. \tag{B1}$$

$L_{meas,\lambda}$ denotes the measured radiance at wavelength $\lambda$ and $L_{noabs,\lambda}$ denotes the simulated radiance without absorption. The parameter $a$ scales with the brightness of the measured spectrum and the parameter $x$ scales with the transmittance and is, therefore, a measure of absorption. Since the transmittance is an exponential function depending on the propagated distance of the light and the absorption coefficient, the parameter $x$ is an exponent.





A threshold of the parameter $x$ is defined by visual inspection to distinguish cloudy from clear sky measurements which is then adapted dynamically depending on the viewing zenith angle of the camera, the solar zenith angle, and the column integrated water vapor density of ECMWF ERA-Interim reanalysis data (Dee et al., 2011). As clouds reflect large amounts of solar radiation they appear as bright features in the measurements. This makes it also useful to define a threshold of the parameter $a$ which has been done by a visual inspection. Additional, the signal to noise ratio, providing information about the reliability of individual measurements, is used to determine the cloud mask. Also all clouds smaller than 3x3 are removed because it was seen that in cases with sun-glint accompanied by either high aerosol concentrations or too high water vapor concentrations in ERA-Interim data, the cloud mask fails by detecting widely scattered and very small clouds.

*Author contributions.* LH was in charge of the presented method, development of the three-dimensional cloud macro- and microphysics retrieval and the manuscript. FG developed and provided the cloudmask. MG derived the used WALES cloud top heights. TK developed the shadow mask, the effective radius retrieval and also provided valuable input during the development and verification of the method. BM and TZ prepared the field campaigns, provided valuable input during the development of the method and contributed to the final version of this paper.

*Competing interests.* The authors declare that they have no conflict of interest.

*Acknowledgements.* The authors thank the German Science Foundation (DFG) for supporting the HALO NARVAL-II and NAWDEX campaigns within the priority program SPP1294 Atmospheric and Earth System Research with the Research Aircraft HALO (High Altitude and Long Range Research Aircraft). The authors acknowledge support from the Deutsche Forschungsgemeinschaft (DFG) through grants ZI 1132/3-1, ZI 1132/4-1, as well as the Max Planck Society, the German Aerospace Center (DLR) and DFG travel funding RA 1400/9-1. Additionally, the authors thank the pilots and appreciate support from the Flugbereitschaft of DLR and *enviscope GmbH* for the integration of specMACS. Lucas Höppler also received funding by subproject B4 of the Transregional Collaborative Research Center SFB / TRR 165 "Waves to Weather" (www.wavestoweather.de) funded by the German Research Foundation (DFG). The author thanks the supporting comments and suggestions of Felix Gödde, Tobias Kölling, Manuel Gutleben, Veronika Pörtge, Bernhard Mayer and Tobias Zinner. Many thanks goes also to Ottavia Balducci for proofreading mathematical formulas.



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
