# Peer review of "Synergy of Active- and Passive Remote Sensing: An Approach to Reconstruct Three-Dimensional Cloud Macro- and Microphysics"

_Atmospheric Measurement Techniques, 2020_

## Referee Comment (RC1) · Anonymous Referee #1 · 9 May 2020

This manuscript reports a method for providing three-dimensional (3D) cloud macro- and microphysical properties, using observations of shortwave spectrometer, microwave radiometer, lidar, radar and dropsondes taken from the German research aircraft HALO. The basis and practical implementations of the methodology, including determinations of cloud boundaries and microphysical cloud profiles, are detailed. Examples are shown using data from the recent field campaign. While this work has not been evaluated against in-situ measurements, the retrieval and 3D reconstruction are checked through intercomparisons between the observed and simulated radiances. As indicated in the Conclusion section, the work can be applied to the future EarthCARE satellite mission, which is exciting and will have great scientific impacts.

[Figure]

I have had high hopes for this manuscript. The authors are thorough about the details and are candid about the limitation of the method, which is greatly appreciated. However, the presentation of the manuscript is disappointing. I must confess that I feel it is a bit unkind to submit the manuscript in the current form. I know this comment may be hard to take, but I hope the authors can understand my frustration. The manuscript lacks a clear logic flow and is disorganized and hard to follow. The material and figures are not carefully chosen. In-depth discussions and insights are lacked in the current form. I really feel it is a missed opportunity, because this is excellent work, but the manuscript didn't do it justice.

Some specific comments –

The introduction lacks a direct, clear focus and logic flow, as we can tell by references that were not introduced in a proper order. We can also tell that the introduction is not right, when the abstract is all about 3D, but the word of "3D" didn't get mentioned until the fifth paragraph. I would think that the uniqueness of the method is about 3D, so perhaps the introduction can focus on 1) the importance of cloud inhomogeneity and 3D information in terms of quantifying cloud radiative effect and understanding the role of clouds in the weather and climate system, 2) why this work is one of the few that can provide 3D cloud fields, and 3) how exactly the advance in these retrievals can help to improve models.

The Discussion and Conclusion section includes some information that is either not found or somewhat inconsistent with the main body of the text. For example, it states that microphysical profiles within the clouds are determined using the HAMP measurements, which can mean quite a few things because HAMP includes passive and active sensors. Indeed, radar reflectivity is passed to non-nadir pixels, but I don't see how it is used in helping constrain cloud microphysical properties. If radar data is only used for detecting multi-layer clouds as described in the manuscript, then radar data is underused and the synergy between passive and active is actually quite minimal. Also, the conclusion section highlights the values of cloud droplet number concentrations, but

this is not mentioned in the main text (although I might miss it). The effective radius profile is said to be constant, but the whole section 3.3 is talking about how to determine effective radius profile. Some clarification and consistency check are necessary.

I understand section 3.3 is one of the key components, but the current material can be found in textbooks so it is better to be summarized in a more concise way. Also, it is more useful and important to provide discussions about the impacts of the assumptions, and the understanding of the accuracy required in the input parameters in order to make useful retrievals. For example, how accurate do cloud base height and cloud top height need to be? How does that affect the capability of the retrieval method in satellite applications in which dropsondes are not available?

The material in section 4 can be condensed, and the flow should be reorganized, i.e., talking about PCA first and then introducing the matching process. In fact, PCA fits better in section 2 when the instrument/data were introduced. Why talking 18 PCA components, and then immediately, it is decided to use 50 components? Additionally, the authors sometimes use "not change much" or "visually no difference", which is not a scientific way to present results because readers cannot replicate. How much is not much, and how small is small? I would suggest describing things more clearly.

In the end, the method seems to aim for providing one profile for one scene. It would be good to have some discussions on the usefulness of such retrievals.

The manuscript needs some serious clean-up. The subtitles are not informative, and their orders didn't make sense. I am afraid that the symbols are a mess. For example, symbol 'a' is used for both radius of cloud droplet and the degree of entrainment. N is used as a constant and as a function of droplet size. Effective radius is sometimes a function of height, and sometimes not (e.g., equation 17).

Figures:

Figure 6: This illustration doesn't add much. It does not provide information on what

determines a good match and a bad match. While the matching process is based on passive radiance spectrum, the illustration seems to show either radar/lidar column information, which can be misleading. Additionally, readers cannot see what is going on for the good match pixel, so I am not sure what this figure can achieve.

Figure 7: I spent a lot of time on reading back and forth, and still cannot figure out what dashed lines represent here, and what the key point is.

Figure 8: Is (a) just a plot of 600m in red and 0 in black? Why does it deserve to be a plot? Shouldn't it be more informative to explain why it is OK to use infrequent drop-sondes with coarse spatial sampling for the entire cloud scene? The physics behind and the impact on retrievals?

Figure 13: Suddenly, figure captions mention a cloud scene, and no further descriptions. Is it the same scene used in other figures? This figure is quite important and interesting but lacks some details. Without that, it is hard to know if the filtering process is too strict, and why it is OK to assume one profile for a 10x10 km area.

---

## Referee Comment (RC2) · Anonymous Referee #2 · 28 Jul 2020

This paper presents a method to infer 3D cloud fields of macro- and micro-physical properties using a synergy of 2D cloud imagery (vis, NIR + microwave), lidar and dropsondes. The method is an adaptation of the Barker et al,. 2011 cloud reconstruction method whereby the vertical structure of the clouds is inferred by extrapolating information from a central track under the assumption that profiles with similar radiances have similar vertical profiles. The method differs from the Barker et al., 2011 approach by using lidar and radiosonde to retrieve cloud top height and cloud base height respectively in the central track, rather than radar reflectivity. The application of the method is also different; measurements from instruments aboard the German research aircraft HALO are used rather than satellite data allowing finer resolution measurements of

cumulus clouds which are difficult to observe from e.g., CloudSat.

The overall scope of the paper is definitely relevant to AMT and has the potential to be a great paper of interest to a broad audience. However, often frustrating to the reviewer, the paper feels unfinished at times, particularly in the results section. The paper has a number of co-authors and it is surprising that some of the quite obvious issues with the paper have not been ironed-out. As a simple example, many of the figure captions need work to be clear; the captions of Fig. 10. and 11 feel rushed and it is not clear if the figures are from the same cloud field. I therefore strongly suggest that the paper is reviewed carefully amongst co-authors before submission of the revised version.

General comments

- The manuscript does not contain sufficient information to reproduce the results in its current form. How many different scenes were used to generate the statistics in Table 1? Is the same scene used in all the Figures?

- How was the radar used in the study? There is a brief mention in the introduction on page 3 line 17 'The radar is used to determine multiple cloud layers', and then is not mentioned again (apart from to define its frequency in Section 2.2). Could the radar be used to define a more accurate cloud base? Could the profiling information of the radar be used to improve the microphysical retrievals?

- The introduction lacks focus, particularly in relation to '3D'. How will 3D cloud fields help reduce uncertainties in cloud processes? What other 3D cloud retrievals are available?

- The abstract makes a subjective claim that the consistency check shows 'good agreement'. Do you think that can be claimed given the disagreements in the radiance histograms i.e. Fig 16c) and 16f)? What would be a good agreement? Does the method outperform a 'dumb' retrieval of assuming a constant cloud base height (perhaps by taking the average cloud top height seen by the lidar for a given cloud scene)

- How justifiable is the assumption of constant cloud base between dropsondes?

- Would an infra-red channel allow cloud-top temperature to be used to better retrieve cloud top height as is done for e.g., MODIS cloud top height retrievals? If an infra-red channel was available on HALO would the method become obsolete? Perhaps I am not understanding something here.

- The discussion of 'shadowing' in the paper could be elaborated. In the introduction you state 'Shadow effects can be eliminated', which led me to think that the method would somehow account for shadowing in the retrieval, but I don't think this is the case-perhaps 'Shadow affected clouds can be removed...' would be more appropriate. How much does solar zenith angle affect the results?

Specific comments

- I do not think that the diagram of specMACS in Fig 1 adds much to the paper. Would it not be more useful to include a schematic/flow diagram of how the method works?

- What does 'flight security' mean on page 5 line 22?

- In Fig.5 Why does the liquid water content for a=0.5 appear more than half of a=1.0? Perhaps it appears that way because the axes do not start at '0'.

- What does 'm' mean in Fig 6?

- What data/measurements are used as input to Fig 7? This figure would be impossible to reproduce without this information!

- What does 'px' mean in the figures? Pixel? How is that a time? Why is the range different in Fig 16. Compared to Fig 18 (0-300 vs 0-1600)?

- Would other statistics., e.g., RMSE, be useful in evaluating the reconstructions in Section 5.5?

- How is the error of 10% in effective radius (page 29 line 12) justified when the previous

paragraph says 20%?

- In Fig 12., wouldn't a shallower sun angle be more appropriate to show an example of the shadow mask? SZA=3.6 degrees is almost overhead, where shadows would be minimized anyway.

- Why isn't the cloud mask applied before reporting the cloud effective radius in Fig 10? It would be useful to see what cloud effective radius is retrieved near cloud edges.
* * *

---

## Author Comment (AC1) · 30 Apr 2021

Dear Referees,

I would like to appreciate the time and the effort you invested in reading and reviewing my very first paper. The comments are inspiring and will improve my publication significantly. In the following, I quoted your comments and tried to answer them as precise as I could.

General Comments

Comment 1: "[...] The manuscript lacks a clear logic flow and is disorganized and hard to follow. The material and figures are not carefully chosen. In-depth discussions and insights are lacked in the current form."

Answer to comment 1: Thank you for this comment. I agree to the lack of a clear logic flow, the disorganization, and the poor readability. This paper starts with an introduction describing the importance to reconstruct three-dimensional clouds and the different remote sensing instruments. The second section presents the instruments which I used in my research. The third section gives an overview of the necessary theoretical background and describes the sub-adiabatic cloud microphysical model. Section 4 shows how the different instruments can be combined and how the data amount can be processed. This section also explains how the three-dimensional macrophysics (subsection 4.1) and the three-dimensional microphysics (subsection 4.2 & subsection 4.3) is reconstructed. I inserted in the new version a new chapter for the quality of the reconstruction (Section 5). I have also adapted figures, and deleted figures that were not containing much information, and extended parts of the discussion.

Specific Comments

Comment 2: "The introduction lacks a direct, clear focus and logic flow, as we can tell by references that were not introduced in a proper order. We can also tell that the introduction is not right, when the abstract is all about 3D, but the word of "3D" didn't get mentioned until the fifth paragraph. I would think that the uniqueness of the method is about 3D, so perhaps the introduction can focus on 1) the importance of cloud inhomogeneity and 3D information in terms of quantifying cloud radiative effect and understanding the role of clouds in the weather and climate system, 2) why this work is one of the few that can provide 3D cloud fields, and 3) how exactly the advance in these retrievals can help to improve models."

Answer to comment 2: Thank you for this comment. Indeed, the focus of my introduction lies in the reconstruction of three-dimensional clouds. I have not seen this

discrepancy until now and have adapted the introduction in the new version of this document. It starts now with the importance of cloud inhomogeneity and 3D information and shows why this work is one of the few that can provide 3D cloud fields. Also, how these retrievals can help to improve models should be clearer now.

Comment 3: "The Discussion and Conclusion section includes some information that is either not found or somewhat inconsistent with the main body of the text. For example, it states that microphysical profiles within the clouds are determined using the HAMP measurements, which can mean quite a few things because HAMP includes passive and active sensors. Indeed, radar reflectivity is passed to non-nadir pixels, but I don't see how it is used in helping constrain cloud microphysical properties. If radar data is only used for detecting multi-layer clouds as described in the manuscript, then radar data is underused and the synergy between passive and active is quite minimal. Also, the conclusion section highlights the values of cloud droplet number concentrations, but this is not mentioned in the main text (although I might miss it). The effective radius profile is said to be constant, but the whole section 3.3 is talking about how to determine effective radius profile. Some clarification and consistency check are necessary."

Answer to comment 3: Only the Liquid Water Path of the HAMP microwave radiometers is used to retrieve the Liquid Water Content profiles within the clouds. The HAMP (cloud) radar reflectivity was only used to identify multiple cloud layers. Unfortunately, available radar data was not suited to determine a, for example, more realistic Cloud Bottom Height due to the limited sensitivity to the small droplets. This information can be found in the conclusion section. It is good to hear that this part should be written more clearly. Now this information can also be found in Section 3. The Cloud Droplet Number Concentration values are mostly a by-product, and yes, it is true that they do not disappear in the main body. I did not include more information since the paper is already very long and difficult to understand. Thank you for the comment. I already have criticised myself concerning this part. I have removed them from the conclusion in the new version. The Cloud Droplet Effective Radius profile is determined using the

sub-adiabatic theory of subsection 3.3 and the filtered Cloud Droplet Effective Radius values determined using Nakajima and King (1990) according to subsection 4.3. Since the filtering method eliminates many Cloud Droplet Effective Radius values, only a small subset can be used as constraints for the sub-adiabatic Cloud Droplet Effective Radius profile. Thus, only one Cloud Droplet Effective Radius profile is used for the considered case study shown in Figure 15. Saying that this profile is constant is not concise – I appreciate this comment. This has also been criticised by the second reviewer. I have adapted it accordingly.

Comment 4: "I understand section 3.3 is one of the key components, but the current material can be found in textbooks, so it is better to be summarized in a more concise way. Also, it is more useful and important to provide discussions about the impacts of the assumptions, and the understanding of the accuracy required in the input parameters in order to make useful retrievals. For example, how accurate do cloud base height and cloud top height need to be? How does that affect the capability of the retrieval method in satellite applications in which dropsondes are not available?"

Answer to comment 4: I have summarized this section a bit. But I think it is very crucial to bring much details in it for readers that have not such a great experience in cloud microphysics. Moreover, I think it is a great idea to include a discussion about the impacts of the assumptions and the accuracy of the input parameters. Your question, "[. . .] how accurate do cloud base height and cloud top height need to be?" is a tricky one. Which accuracy do we want in the first place? My first naive answer would be: as accurate as possible. In the case of satellite applications, one would need to estimate the Cloud Bottom Height from radiosonde profiles or model data. This should be investigated in a further study.

Comment 5: "The material in section 4 can be condensed, and the flow should be re-organized, i.e., talking about PCA first and then introducing the matching process. In fact, PCA fits better in section 2 when the instrument/data were introduced. Why talking 18 PCA components, and then immediately, it is decided to use 50 components?

Additionally, the authors sometimes use "not change much" or "visually no difference", which is not a scientific way to present results because readers cannot replicate. How much is not much, and how small is small? I would suggest describing things more clearly."

Answer to comment 5: Thank you for your comment. It is, for me, not logical to introduce first the PCA and then the matching process, because one must first understand the matching process, since this process is made faster with the PCA. I have written this part now clearer in the new version of the document.

Comment 6: "In the end, the method seems to aim for providing one profile for one scene. It would be good to have some discussions on the usefulness of such retrievals."

Answer to comment 6: The method aims to provide one Cloud Droplet Effective Radius profile for one scene since only a few retrieved Cloud Droplet Effective Radius values can be used to fit the sub-adiabatic model. However, for every atmospheric column (here with 15 m resolution) a Liquid Water Content profile is retrieved because the Liquid Water Path retrieved by HAMP is not or not much influenced by three-dimensional radiative effects (compared to the Cloud Droplet Effective Radius retrieval). It would be beneficial if more data, e.g. the Cloud Droplet Effective Radius from cloud bow or glory retrievals, are integrated into this method in the future. This data integration would make this method unique.

Comment 7: "The manuscript needs some serious clean-up. The subtitles are not informative, and their orders didn't make sense. I am afraid that the symbols are a mess. For example, symbol 'a' is used for both radius of cloud droplet and the degree of entrainment. N is used as a constant and as a function of droplet size. Effective radius is sometimes a function of height, and sometimes not (e.g., equation 17)."

Answer to comment 7: I have given better subtitles in the new version. I ordered them more appropriately and renamed the symbols.

Comment to figure 6: "This illustration doesn't add much. It does not provide information on what determines a good match and a bad match. While the matching process is based on passive radiance spectrum, the illustration seems to show either radar/lidar column information, which can be misleading. Additionally, readers cannot see what is going on for the good match pixel, so I am not sure what this figure can achieve."

The answer of the comment to figure 6: I neither find three-dimensional figures very attractive since most of them do not show valuable information. However, as an overview of the method, I do like this image a lot, and sometimes a picture is worth more than a thousand words. In my opinion, it makes it much easier to understand the aim of this method, which is to transfer the nadir measured active or passive remote sensing information on the broader field-of-view of an imager spectrometer. Moreover, the reader must understand where the axes are, the spatial axis $j$ and the temporal axis $i$ (off-nadir pixel) and $m$ (nadir pixel). I have changed the caption of the new version. When the reader is reading the text, I think it should be possible to understand that passive radiance spectra are compared to each other and in case of a small deviation, the measured nadir information is used for the off-nadir location.

Comment to figure 7: "I spent a lot of time on reading back and forth, and still cannot figure out what dashed lines represent here, and what the key point is."

The answer of the comment to figure 7: This picture shows the shortwave infrared spectrum of specMACS, which ranges from about 1050 to 2500 nm. The red dashed lines show the spectrum where the measured Liquid Water Path is the same, but the measured Cloud Top Height is different. The critical point in this figure is to show that various macro- and microphysical properties of the clouds lead to a diverse measured spectrum. This fact is essential for the matching method, which would not work well when, for example, different measured Liquid Water Path values lead to the same measured spectrum. I have deleted the figure in the new version.

Comment to figure 8: "Is (a) just a plot of 600m in red and 0 in black? Why does it

deserve to be a plot? Shouldn't it be more informative to explain why it is OK to use infrequent dropsondes with coarse spatial sampling for the entire cloud scene? The physics behind and the impact on retrievals?"

The answer of the comment to figure 8: I have excluded the plot. Moreover, I have calculated the standard deviation of calculated cloud top height from all dropsonds of that day which can be found now in the new section 3.1. Also, an explanation why it is OK to use them can be found there. I do not understand your comment "the impact on retrievals".

Comment to figure 13: "Suddenly, figure captions mention a cloud scene, and no further descriptions. Is it the same scene used in other figures? This figure is quite important and interesting but lacks some details. Without that, it is hard to know if the filtering process is too strict, and why it is OK to assume one profile for a 10x10 km area."

The answer of the comment to figure 13: Pardon! I have included more information now in the caption. Also, now, always the same cloud scene is shown in every figure.

I once again thank you for all the constructive comments.

Please also note the supplement to this comment:
https://amt.copernicus.org/preprints/amt-2020-49/amt-2020-49-AC1-supplement.pdf

---

## Author Comment (AC2) · 30 Apr 2021

Dear Referees,

I would like to appreciate the time and the effort you invested in reading and reviewing my very first paper. The comments are inspiring and will improve my publication significantly. In the following, I quoted your comments and tried to answer them as precise as I could.

General Comments Comment 1: "[. . .] However, often frustrating to the reviewer, the paper feels unfinished at times, particularly in the results section. The paper has a

number of co-authors and it is surprising that some of the quite obvious issues with the paper have not been ironed-out. As a simple example, many of the figure captions need work to be clear; the captions of Fig. 10. and 11 feel rushed and it is not clear if the figures are from the same cloud field."

Answer to comment 1: Thank you for your comment. I have changed the captions of Fig. 10 and Fig 11. They can be found in Figure 6 of the new version.

Comment 2: "The manuscript does not contain sufficient information to reproduce the results in its current form. How many different scenes were used to generate the statistics in Table 1? Is the same scene used in all the Figures?"

Answer to comment 2: Table 1 only refers to the case study, which is shown in figure 15. This reference should be made also be more concise. I have improved the information.

Comment 3: "How was the radar used in the study? There is a brief mention in the introduction on page 3 line 17 'The radar is used to determine multiple cloud layers', and then is not mentioned again (apart from to define its frequency in Section 2.2). Could the radar be used to define a more accurate cloud base? Could the profiling information of the radar be used to improve the microphysical retrievals?"

Answer to comment 3: My apology. I have not properly introduced the radar itself and will do it in the next version. As I wrote in the conclusion section, unfortunately, the cloud radar was not suited to determine a more realistic cloud bottom height due to the limited sensitivity to the small droplets. If the radar would be adjusted appropriately and build better, it might be used to define a more accurate cloud base and could be used to improve the microphysical retrievals. Sadly, I could not use the radar for more than just determining multiple cloud layers. I hope in the future campaigns the radar information is more suitable. I have adapted the information written in the publication.

Comment 4: "The introduction lacks focus, particularly in relation to '3D'. How will 3D cloud fields help reduce uncertainties in cloud processes? What other 3D cloud

retrievals are available? Answer to comment 4: I appreciate this comment. The first reviewer also suggested this and I have changed the introduction accordingly which focus now more on 3D."

Comment 5: "The abstract makes a subjective claim that the consistency check shows 'good agreement'. Do you think that can be claimed given the disagreements in the radiance histograms i.e. Fig 16c) and 16f)? What would be a good agreement? Does the method outperform a 'dumb' retrieval of assuming a constant cloud base height (perhaps by taking the average cloud top height seen by the lidar for a given cloud scene)?"

Answer to comment 5: I have introduced a plane-parallel 3D cloud field in section 5.3 in the new version of this document as you suggested. I also compared now the developed retrieval to the "dump" retrieval.

Comment 6: "How justifiable is the assumption of constant cloud base between drop-sondes? Answer to comment 6: Over the ocean, the temperature and dew points vary not too much over large areas, which causes a widely constant Cloud Bottom Height. For example, figure 2 of Li, J. M., et al. (2013) shows a mostly stable Cloud Bottom Height for maritime boundary layer clouds for a larger area. I have discussed this issue shortly in the new section 3.1. The mean absolute CBH difference between all drop-sonds on the day I focus in the paper was 59 m. Since I interpolate linearly between two calculated CBH, the error should be smaller."

Comment 7: "Would an infra-red channel allow cloud-top temperature to be used to better retrieve cloud top height as is done for e.g., MODIS cloud top height retrievals? If an infra-red channel was available on HALO would the method become obsolete? Perhaps I am not understanding something here."

Answer to comment 7: I have no experience in this cloud-top temperature retrieval. When there would be an infra-red channel available, and the Cloud Top Height is re-trieved using this channel, then the quality of this retrieval needs to be compared and

evaluated first. This comparation and evaluation could be done using active sensors such as the lidar. In the end, a combination of this retrieval with the retrieval presented in this paper could be a good complement. I have not seen any CTH retrieved from a thermal infrared imager, but I have included your comment in the discussion section of the new version.

Comment 8: "The discussion of 'shadowing' in the paper could be elaborated. In the introduction you state 'Shadow effects can be eliminated', which led me to think that the method would somehow account for shadowing in the retrieval, but I don't think this is the case - perhaps 'Shadow affected clouds can be removed. . .' would be more appropriate. How much does solar zenith angle affect the results?"

Answer to comment 8: Thank you for this comment. I have written now "shadow affected cloud parts can be removed."

Specific Comments

Comment 9: "I do not think that the diagram of specMACS in Fig 1 adds much to the paper. Would it not be more useful to include a schematic/flow diagram of how the method works?"

Answer to comment 9: I agree with that comment. It only shows how it looks from the outside and where the cameras are located. I think a reference to the publication of Ewald et al. (2013) is sufficient. I have removed the figure.

Comment 10: "What does 'flight security' mean on page 5 line 22?"

Answer to comment 10: Dropsonds are thrown from the aircraft into a region where other passenger aircrafts fly. Flight security means that the dropsonde can only be thrown from the aircraft if there are no other objects in the airspace. This is meant by "flight security". I have included it into the new version.

Comment 11: "In Fig.5 Why does the liquid water content for a=0.5 appear more than half of a=1.0? Perhaps it appears that way because the axes do not start at '0'."

Answer to comment 11: Thank you for this comment. This is a typo and should be a=0.6. I am sorry for that. I have corrected it.

Comment 12: "What does 'm' mean in Fig 6?"

Answer to comment 12: The index m indicates the temporal axis of the nadir pixels. I have written it more precisely in the new version.   Comment 13: "What data/measurements are used as input to Fig 7? This figure would be impossible to reproduce without this information!"

Answer to comment 13: These are data measured on the nadir track. Five different measured spectra were used. Three spectra, where the WALES CTH was constant and the LWP varied. Two spectra were taken where a constant LWP was measured, but different WALES CTHs. Thank you for this comment, I got also comments that this figure does not give any information at all. Thus, I have removed it.

Comment 14: "What does 'px' mean in the figures? Pixel? How is that a time? Why is the range different in Fig 16. Compared to Fig 18 (0-300 vs 0-1600)?"

Answer to comment 14: Thank you for this comment. Pixels indicate the size of the image. Indeed, it is not a very good quantity for time. I have adapted the axis in the new version.

Comment 15: "Would other statistics., e.g., RMSE, be useful in evaluating the reconstructions in Section 5.5? How is the error of 10% in effective radius (page 29 line 12) justified when the previous paragraph says 20%?"

Answer to comment 15: What might be more interesting might be to compare the retrieval with a "dump" two-dimensional one and estimate the differences, as you and I have written above. I have decided to use the Mean Average Error. An error of 10 % in Cloud Droplet Effective Radius is not justified when the previous paragraph says 20 %. I have improved the new version accordingly

Comment 16: "In Fig 12., wouldn't a shallower sun angle be more appropriate to show

an example of the shadow mask? SZA=3.6 degrees is almost overhead, where shadows would be minimized anyway."

Answer to comment 16: A shallower sun angle would be more appropriate. I have added the scene I used in Figure 6 now everywhere. The solar zenith angle is in this case about 21.5 degree.

Comment 17: "Why isn't the cloud mask applied before reporting the cloud effective radius in Fig 10? It would be useful to see what cloud effective radius is retrieved near cloud edges."

Answer to comment 17: I got several comments that I should show the whole scene without the cloud mask in conferences. I have applied the cloud mask now for the Cloud Droplet Effective Radius (Figure 6 now).

I hope I answered all your comments and thank you once again for all the constructive comments.

References: Li, J. M., et al. "A new approach to retrieve cloud base height of marine boundary layer clouds." Geophysical research letters 40.16 (2013): 4448-4453.

Please also note the supplement to this comment:
https://amt.copernicus.org/preprints/amt-2020-49/amt-2020-49-AC2-supplement.pdf